# Decreased Systemic Monocyte Colony Protein-1 (MCP-1) Levels and Reduced sCD14 Levels in Curcumin-Treated Patients with Moderate Anxiety: A Pilot Study

**DOI:** 10.3390/antiox13091052

**Published:** 2024-08-29

**Authors:** José Joaquín Merino, José María Parmigiani-Cabaña, José María Parmigiani-Izquierdo, Rubén Fernández-García, María Eugenia Cabaña-Muñoz

**Affiliations:** 1Facultad de Farmacia, Departamento de Farmacología, Farmacognosia y Botánica, Universidad Complutense de Madrid (UCM), 28040 Madrid, Spain; 2Centro de Rehabilitación Oral Multidisciplinaria, 30001 Murcia, Spain; jmpc@clinicacirom.com (J.M.P.-C.); jmparmi@clinicacirom.com (J.M.P.-I.); mecjj@clinicacirom.com (M.E.C.-M.); 3Department of Nursing, Physiotherapy and Medicine, University of Almeria, 04120 Almeria, Spain; rubenfer@ual.es

**Keywords:** anxiety, inflammation, chemokines, curcumin, monocytes, sCD14, oxidative stress, antioxidants, MCP-1, cortisol, CCR2, nutraceuticals, stress, anxiety, depression, behavior, inflammation, neuroprotection, neuropsychiatric disorders, neurology

## Abstract

Psychosocial stress may alter cortisol and/or affect the normal functioning of the immune system. Curcuminoids can promote beneficial effects in neuropsychiatric diseases. We evaluated whether curcumin supplementation for 15 consecutive days (1800 mg/day) would decrease systemic MCP-1, sCD14, and TNF alpha levels in patients with moderate anxiety (*n* = 81). A total number of 81 subjects were enrolled in this study, divided into the following groups according to their Hamilton scores: a control group including patients without anxiety who were not taking curcumin (Cont, *n* = 22) and an anxiety group including patients with moderate anxiety (Anx, *n* = 22). The curcumin-treated patients experienced moderate anxiety, and they take curcumin for 15 consecutive days (Anx-Cur (after), *n* = 15, 1800 mg/day). An evaluation of 128 patients was conducted, which allowed for their assignment to the study groups according to their scores on Hamilton scale II. The cortisol levels were quantified in salivary samples through ELISA (ng/mL), and malonaldehyde (MDA) levels were measured in plasma via the TBARS assay as an index of lipoperoxidation. Several systemic proinflammatory cytokines (pg/mL: MCP-1, TNF alpha, IL-1 beta) and mediators were quantified through ELISA (pg/mL), including systemic sCD14 levels as a marker of monocyte activation. A two-way bifactorial ANOVA was conducted to evaluate the contributions of the anxiety factor (Anx) and/or curcumin factor (Cur) in all the tested markers, including interactions between both factors. High systemic MCP-1 and elevated sCD14 levels were observed in patients with moderate anxiety, which were reduced with curcumin supplementation. In addition, curcumin prevented cortisol overexpression and decreased MDA levels as an antioxidant response in these patients. Collectively, curcumin presented anti-chemotactic effects by reducing systemic MCP-1 levels in anxiety. Curcumin decreased systemic MCP-1 as well as sCD14 levels in patients with moderate anxiety.

## 1. Introduction

Psychosocial stress in patients has been associated with high cortisol levels in saliva and plasma, which can contribute to the detrimental effects of chronic stress, anxiety [1], and other neuropsychiatric diseases [2,3,4]. The hypersecretion of cortisol can affect the immune system [5] and, over time, can cause the hypothalamic–pituitary–adrenal (HPA) axis to become overactive [6,7,8,9], provoke functional alterations, and/or accelerate the progression of certain neuropsychiatric disorders [3]. Proinflammatory cytokines, such as TNF alpha, provoke disease-related behaviors and accelerate the progression of age-related disorders [10]. In fact, the innate immune system was activated with a single injection of proinflammatory cytokines (IL-1β or TNF-α) in the rat hippocampus, leading to HPA axis overactivation [11]. This activation of the HPA axis contributes to disease-related behavior [12] and other behaviors [13]. The topic of interpersonal reactivity has been extensively studied in recent years. Diotaiuti et al. (2021) analyzed the validity of the Interpersonal Reactivity Index metric, highlighting its importance in understanding social dynamics [13].

On the other hand, glucocorticoids promote oxidative stress and increase the release of proinflammatory cytokines (e.g., IL-1β, TNF-α, prostaglandins) [11]. Stress can induce the recruitment of monocytes into different tissues in the context of neuropsychiatric diseases [14]. In addition, human monocytes are a heterogeneous cell population of peripheral “proinflammatory monocytes” [15]. In particular, the monocyte chemoattractant protein-1 chemokine (MCP-1, also termed CCL2), which binds to the CCR2 chemokine receptor, recruits mononuclear cell infiltrates into the brain [16].

In this context, certain active principles from plants, such as curcuminoids, have health-promoting effects in humans. Curcuminoids derived from the natural rhizome of *Curcuma longa* (turmeric root) are lipophilic polyphenols that are rich in bisdemetoxicurcumin and demerotoxicurcumin compounds [17]. Curcumin presents antioxidant, anti-inflammatory, cardiovascular, and neuroprotective effects in humans. However, the efficacy of curcumin depends on its bioavailability, dose, and the state of progression of the disease [18,19,20]. In several pre-clinical studies, curcumin was able to increase the lifespan of individuals by inducing anti-inflammatory effects via activation of the transcription factors p65 NF-kappa beta or Nrf-2, and enhanced endogenous antioxidant enzymes (e.g., superoxide dismutase, SOD-1), acting as protective mechanisms [21].

This study hypothesized that anxiety and/or curcumin could significantly influence cortisol and systemic inflammatory cytokine levels in patients with moderate anxiety (MCP-1, sCD14, TNF alpha).

## 2. Aim

This study aimed to evaluate whether curcumin supplementation for 15 consecutive days (1800 mg/day) could reduce systemic MCP-1, sCD14, and TNF alpha levels in patients with moderate anxiety.

## 3. Methods

### 3.1. Study Design

Initially, 128 subjects were assessed for their Hamilton score and were divided into four study groups according to these scores. This randomized pilot study enrolled a total of 81 patients. We enrolled 44 patients with moderate anxiety; half of them (*n* = 22) received a placebo (anxiety group, Anx; without taking curcumin), and 22 patients were supplemented with curcumin for 15 consecutive days (Anx-Cur (after) group: 1800 mg twice a day, *n* = 22). The controls were subjects without anxiety (control, *n* = 15) and curcumin-treated patients without anxiety (Cur, *n* = 15). All patients were randomly divided into study groups according to their Hamilton scores (see the enrollment process depicted in Figure 1).

All patients supplemented with curcumin received a commercial powdered phytosome formulation—Merive@ with phosphatydilserine (Merive@)—for 15 consecutive days (1800 mg twice a day, from 8.00–10.00 a.m.). Merive@ curcumin contains curcuminoids (desmethoxycurcumin and bisdesmethoxycurcumin) of 95% purity, and after treatment, levels of curcuminoids were detected in plasma and tissues [22,23,24,25].

As an additional control, we compared all evaluated biomarkers in curcumin-treated patients with moderate anxiety after 15 days of taking curcumin [Anx-Cur (After) group, *n* = 22, 1800 mg twice a day], compared with their own basal levels [before taking curcumin: Anx-Cur (Before) group, *n* = 22]. All obtained results were blinded to the physicians and researchers until the end of the study.

Hamilton scores, cortisol, several systemic proinflammatory markers (pg/mL: TNF Alpha, MCP-1), and sCD14 were compared between the study groups. Hamilton and salivary cortisol levels (ng/mL) [26,27] were measured, and blood samples were collected on the first day (day 1: visit to the clinic) as well as after 15 days of curcumin supplementation (day 15).

The average age of subjects was 44 for controls and 46 for patients with moderate anxiety. The Interpersonal Reactivity Index (IRI), in its brief Italian version, was validated by Diotaiuti et al. (2021) in terms of its effectiveness and reliability [13]. In this study, we used the Hamilton scale II for the evaluation of anxiety in all participants. They filled in the Hamilton scale II questionnaire by evaluating 14 different items not only on their initial visit to the Clinic (CIROM: Centro de Implantología, Rehabilitación Oral Multidisciplinaria, Murcia, Spain) but also on the last day of curcumin supplementation (day 15). CIROM has been certified by AENOR, Rule UNE 179001, and ISO Rules 9001, which provides a guarantee of quality.

Peripheral blood samples were collected in EDTA tubes by a healthcare professional, and plasma was isolated after centrifugation at 2780 rpm for 10 min at 4 °C; the supernatant was stored at 80 °C for further quantification of systemic proinflammatory mediators via ELISA (pg/mL: TNF-alpha, MCP-1) as well as sCD14 (pg/mL: a marker of monocyte activation).

On the last day of oral curcumin supplementation, all patients filled in the Hamilton scale II again, including controls. The diet of the patients was periodically supervised. All enrolled participants were properly instructed before taking these supplements, and they signed the appropriate consent paperwork. All efforts were made to minimize the number of patients. We assert that all procedures contributing to this work were in compliance with the ethical standards of the Declaration of Helsinki of 1975, as revised in 2008 (updated in 2000). Additionally, this study complied with relevant ethical standards and was institutionally approved by the committee of the CIROM Clinic (# 052016), as well as PI-1032. 

Two-way ANOVA was performed to evaluate the effects of the moderate anxiety factor (¨Anx factor¨), curcumin supplementation (¨cur factor¨), and possible interactive effects between both factors (¨Anx factor¨ * ¨cur factor¨) for all systemic markers tested in our study. The scheme in Figure 1 depicts the enrollment of patients (see also Appendix A).

The powdered curcumin formulation (Meriva@) is a curcumin phytosome phosphatidylcholine preparation composed of available phospholipid–phytochemical forms with good gastrointestinal absorption. The water solubility of the phytochemicals is enhanced due to the amphipathic properties of the phospholipid, thus increasing the bioavailability of curcuminoids [23]. In rats, the peak plasma concentration was 5-fold higher for Meriva@-treated animals (the combination of curcumin phospholipids used) when compared to unbound curcumin forms [22,23]. Meriva^®^ (phytosome curcumin) showed no pharmacological interactions with certain drugs (e.g., antiplatelet agents, anticoagulants, and thyroid replacement therapy), at least at dosages routinely used for nutritional supplementation of patients. 

### 3.2. Clinical Characterization of Patients 

All enrolled patients had a medium/high sociocultural status, 75% of them had completed their primary high school degree, and 68% had a bachelor’s degree. They had not taken antioxidants during the six months prior to the beginning of this pilot study. All enrolled participants visited the CIROM Clinic (Centro de Rehabilitación Oral Multidisciplinaria, Murcia), a private dental company. On the first day (visit to the Clinic), all patients filled in the Hamilton scale II and plasma samples were collected in EDTA tubes, including control subjects. On the first and last days of curcumin supplementation, plasma was collected by an authorized healthcare professional via blood extraction and deposited into 5 mL EDTA tubes, and salivary samples were collected for cortisol quantification (ng/mL) via ELISA. Plasma was collected between 8 and 10 a.m. for systemic quantification of biomarkers (MCP-1, sCD14, MDA) as well as salivary cortisol levels (ng/mL) in order to avoid the influence of circadian fluctuations on cortisol levels (see Table 1).

### 3.3. Inclusion Criteria

This study followed the Declaration of Helsinki (1974, and updated in 2000), and all enrolled healthy subjects were properly instructed before taking curcumin and randomized into study groups. They signed the appropriate consent paperwork, and efforts were made to minimize the number of patients and ensure their anonymity. 

The 128 initial participants filled in the Hamilton scale II, and 81 of them were enrolled according to their obtained scores. Other scales, such as the Interpersonal Reactivity Index (IRI), have also been validated and proven to be effective and reliable for behavioral analysis [13], which is consistent with the analysis of the Hamilton scale in our study. The patients were 38–50 years old (average age, also for controls). All enrolled subjects had a normal BMI, as obesity/metabolic syndrome can increase proinflammatory markers due to increased MCP-1 protein levels under obesity [24]. Thus, we enrolled patients within the normal BMI range. The body mass index (BMI) is measured using the formula of weight in kilograms divided by the height in meters squared (kg/m^2^). A BMI less than 24.9 kg/m^2^ is considered lean, values from 25 to 29.9 kg/m^2^ are associated with overweight, and a BMI of 30 kg/m^2^ or higher is associated with obesity.

### 3.4. Exclusion Criteria

We excluded patients with obesity, metabolic disorders (e.g., thyroid disease, diabetes, Cushing syndrome), inflammatory/autoimmune conditions, skin diseases, oral disease (e.g., gingivitis, periodontitis), and/or neuropsychiatric diseases (e.g., depression, schizophrenia). Patients with allergies or intolerances to the active ingredients or excipients were also excluded. None of the subjects had received immunosuppressive drugs, nutritional supplementation with antioxidants, or vaccination for at least 6 months prior to their inclusion in this study. In addition, those with a diet rich in fat were not considered, given their low chronic grade of inflammation; additionally, subjects with blood diseases, digestive problems, autoimmune diseases, hypertension, dermatological problems, or kidney/urological diseases were not considered. Finally, women experiencing menopause, taking corticoids, receiving immunomodulatory drugs, or suffering from COVID-19 were also excluded. Participants who did not meet the eligibility criteria or those with a lack of adherence to the treatment were not considered (*n* = 2, see Appendix A).

### 3.5. Hamilton Scale Type II: Evaluation of Anxiety

The Hamilton scale type II evaluates the severity of stress/anxiety symptoms in patients through the evaluation of 14 different items. This scale evaluates psychological distress and mental agitation as anxiety markers, as well as somatic anxiety. Although it is still widely used as an outcome measure in clinical trials, it has been criticized for its occasionally poor ability to discriminate between anxiolytic and antidepressant side effects vs. somatic anxiety. Despite this feature, the reported levels of inter-rater reliability for the scale are acceptable. The scale has been translated into Cantonese (for China), French, and Spanish [25]. In Hamilton scale II, each item is scored on a scale of 0 (not present) to 4 (severe), with a total score range of 0–56, where a score less than 17 indicates mild severity, scores between 18 and 24 are associated with mild to moderate severity, and scores of 25–30 indicate moderate to severe anxiety in patients [25]. 

### 3.6. Thiobarbituric Acid Assay (TBARS): Malondialdehyde (MDA) Levels as an Index of Lipid Peroxidation 

Malondialdehyde (MDA) levels were quantified following a modified procedure based on the protocol of Tiwari and Chopra (2011) [26]. During the lipoperoxidation process, the formation of 4-hydroxinonenal promotes arachidonic acid oxidation, which reacts with proteins. A standard curve was created using a gradient of thiobarbituric acid (TBARS) concentrations and heating the TBARS reagent to 100 °C. Then, 0.5 mL of TBARS was added to the upper fraction of all plasma samples. The heated reagent was added to a plate in TBA (pH 6.8) with 0.5 mL Tris-HCl buffer. All plasma samples were incubated for 2 h at room temperature (R.T.) before adding 1 mL of TBA (0.67% for 10 min). As soon as all samples were cold, 0.5 mL of distilled water was added to the plate, which was finally measured at 532 nm. The TBA concentration was calculated via interpolation with the standard curve, and MDA levels were expressed as mg of MDA (MDA/mg protein). The TBARS assay shows the TBA content in the study groups as a percentage of that in the control, which is 100% [26].

### 3.7. Quantification of Cortisol via ELISA in Saliva (ng/mL)

In recent years, salivary cortisol detection has been considered an index of indirect activation of the hypothalamic–pituitary–adrenal (HPA) axis, given that its detection via ELISA is a reproducible method. The release of cortisol is rhythmic and influenced by the sleep–wake cycle. Therefore, we quantified salivary cortisol levels obtained at 8.00–10.00 a.m. [27], given the demonstrated correlation between blood cortisol concentration and salivary levels [28]. Salivary cortisol during the diurnal cycle typically ranges from 0.5 to 0.05 µg/dL [27]. In the morning, the mean salivary cortisol range is 3.6–8.3 nmol/L, while it decreases at night (ranging from 2.95 to 2.1 nmol/L) [28]. Likewise, variations in cortisol throughout the day can be considered indicators of high perceived stress [28,29,30].

### 3.8. ELISA Method for Systemic MCP-1, TNF Alpha, and sCD14 Protein Levels (pg/mL)

Systemic MCP-1 (Monocyte colony protein-1, also termed CCL2; Leti, Spain) and sCD14 (a marker of monocytes; R&D, Chicago, IL, USA) levels were quantified via ELISA in plasma samples following the manufacturer’s instructions (Leti, Barcelona, Spain) and my own protocols (Merino et al. [31]). Standards and samples were loaded onto plates, and plasmas were incubated overnight (o/n) in a humidified chamber.

After 3 washes with washing buffer (PBS: phosphate-buffered saline, 0.02% Tween-20, and thimerosal), a polyclonal mAb specific to MCP-1, sCD14, or TNF alpha was loaded onto a microplate for 2 h at 37 °C (100 μL MCP-1 of 100 μL for sCD14, 50 μL of TNF primary alpha antibody). Following 5 washes to remove the unbound antibody–enzyme reagent, non-specific binding was halted using a BSA blocking buffer for 30 min at room temperature in the dark. After 5 washes, 100 μL of diluted horseradish peroxidase (HRP)-conjugated anti-rabbit secondary IgG was incubated for 30 min at room temperature (25 °C) for each antibody. After 3 washes with washing buffer, 100 μL of 0.002 mol/L ortho-phenylendiamine dihydrochloride (OPD) plus 3% hydrogen peroxide (H_2_O_2_) was added to all wells in citrate buffer at 4.5 pH for 10 min (R.T), following in-house protocols [31]. The colorimetric reaction was stopped by adding 50 μL of 2.5 N of sulfuric acid (H_2_SO_4_) to the plate. Finally, the absorbance was measured at 450 nm in a spectrophotometer (Thermo Scientific; Madrid, Spain) for MCP-1 identification and 520 nm for sCD14 identification. The amount of MCP-1, sCD14, or TNF-alpha (Leti Laboratories) was calculated through interpolation of their respective standard curves. For sCD14 (Leti Laboratories), MCP-1 (also called CCL-2), and TNF alpha, all protein levels were expressed as pg/mL. 

### 3.9. Statistical Analysis 

Normality of data distributions was assessed using the Shapiro–Wilk or Levene test. Based on these results, a two-way ANOVA and a Mann–Whitney U-test, followed by post-hoc Bonferroni or Dunnet’s tests, were applied after the appropriate correction for multiple comparisons among study groups. The bifactorial analysis evaluated the effects of the anxiety factor (¨Anx factor¨ in graphs), curcumin treatment (¨cur factor¨), or their interaction (¨Anx factor¨ * ¨Cur factor¨). The results are expressed as mean ± SEM (standard error media: variance/square root); see graphs 1–5 for cortisol, MDA, MCP-1, sCD14, and TNF alpha levels. The linear relationship between two continuous variables was assessed through the Spearman or Pearson correlation coefficient (*r*). A *p*-value of less than 0.05 is considered statistically significant and strongly significant if *p* is less than 0.001. The Spearman r was used to evaluate correlations between inflammatory cytokines (TNF alpha, MCP-1), systemic sCD14 activation as a monocyte marker, or cortisol, and scores obtained in the Hamilton scale II by study group.

## 4. Methodology

### 4.1. Quantification of Anxiety Levels in Subjects: Hamilton Scale II Items

The Hamilton scale type II evaluates the severity of symptoms of stress/anxiety in subjects by testing 14 different items with different range scores [29,30].

### 4.2. Cortisol Levels in Saliva (ng/mL)

Cortisol levels were quantified through ELISA of salivary samples, following the manufacturer’s instructions [28,29,30,31].

### 4.3. Thiobarbituric Acid Assay (TBARS) for Malondialdehyde (MDA) Quantification as an Index of Lipoperoxidation

The peroxidation of lipids was estimated via a TBARS assay in plasma following the procedure described by Tiwari and Chopra (2011; for further details, see [26]). 

## 5. Results

### 5.1. ELISA for MCP-1, TNF Alpha, and sCD14 Protein Levels in Plasma (pg/mL)

TNF alpha (Leti, Barcelona, Spain), Monocyte colony protein-1 MCP-1 (Leti, Barcelona, Spain), and sCD14 (R&D Systems USA) were quantified through ELISA, following the manufacturer’s instructions and in-house protocols (for details see [27], Table 2).

### 5.2. Bifactorial ANOVA: Hamilton Scores in Scale II

There was a significant effect of the anxiety factor [Anx; F (1,77) = 12.3, *p* < 0.05], the curcumin factor [Cur; F (1,77) = 19.0, *p* < 0.05], and the interaction of both factors [Anx * Cur; F (1,77) = 18.2, *p* < 0.05]; see Figure 2.

The post-hoc analysis confirmed lower Hamilton scores after 15 days of curcumin supplementation [Anx-Cur (After)] compared to basal levels before taking curcumin [Anx-Cur (Before); *p* < 0.05].

### 5.3. Cortisol Levels (Saliva, ng/mL)

Salivary cortisol is a potential biomarker for serum-free cortisol in plasma [32,33]. Significant effects were observed for the curcumin factor [F (1,76) = 0.108, *p* < 0.05], the anxiety factor [F (1,77) = 108.2; *p* < 0.05], and the interaction of both factors [Anx * Cur; F (1,77) = 7.45, *p* < 0.05].

The post-hoc analysis also confirmed decreased cortisol levels in curcumin-treated patients with moderate anxiety compared to control subjects (without anxiety), with a *p*-value of less than 0.05, indicating a statistically significant effect. The Holm–Sidak post-hoc method also confirmed reduced cortisol levels in curcumin-treated patients with anxiety compared to patients with anxiety not taking curcumin (*p* < 0.05). Additionally, after 15 days of supplementation, decreased cortisol levels were observed in the Anx-Cur group (After) group, with respect to their own basal levels before taking curcumin [Anx-Cur (Before)], with a statistically significant effect (see Figure 2).

### 5.4. Thiobarbituric Acid Assay (TBARS): Assay for Malondialdehyde (MDA) Quantification as an Index of Lipoperoxidation in Plasma

Increased MDA levels (lipoperoxides) were detected in patients with anxiety [Anx-Cur (After)] compared to healthy subjects (without anxiety, *p* < 0.05), while curcumin prevented this MDA overexpression with respect to basal levels before taking curcumin [Anx-Cur (Before), *p* < 0.05] (see Figure 3).

### 5.5. TNF Alpha

There was a lack of effect for both the anxiety factor [F (1,70) = 1.13, *p* = 0.28, *p* > 0.05, n.s] and curcumin factor [F (1,70) = 1.29; *p* = 0.029, n.s]; thus, we also failed to find a significant interaction effect for these factors [Anx * Cur; F (1,70) = 0.12, *p* = 0.7; n.s] (see Figure 3).

We also observed a Spearman correlation between systemic MCP-1/TNF alpha in curcumin-treated patients and moderate anxiety [Anx-Cur (After), *p* < 0.05].

### 5.6. Monocyte Colony Protein-1 (MCP-1)

We found a statistically significant effect not only for the curcumin factor [F (1,77) = 9.92, *p* < 0.05] but also for the anxiety factor [F (1,77) = 7.78; *p* < 0.5]. In addition, there was a significant interaction effect of the factors [F (1,77) = 5.58, *p* < 0.05] (see Figure 4).

The Spearman correlation revealed a significant relationship between systemic MCP-1 and TNF Alpha in the Anx-Cur (After) group (r = 0.5, *p* = 0.019). In addition, the Spearman coefficient indicated a negative correlation between systemic MCP-1 and sCD14 levels (r = −0.53, *p* = 0.012), while MCP-1 levels were correlated with Hamilton scores in patients with moderate anxiety (r = 0.68, *p* = 0.0003).

In addition, patients with moderate anxiety not taking curcumin [Anx-Cur (Before)] presented a negative correlation between MCP-1 and sCD14 levels (r = −0.46, *p* = 0.031).

### 5.7. sCD14 Levels (pg/mL)

There was a significant effect for not only the curcumin factor [F (1,77) = 8.57, *p* < 0.05] but also the anxiety factor [F (1,77) = 14.45; *p* < 0.5] as well as an evident interactive effect between these factors with regard to systemic sCD14 levels [Anx * Cur; F (1,77) = 7.45, *p* < 0.05] (see Figure 4).

In addition, there was a correlation between systemic sCD14 levels in curcumin-treated patients [Anx-Cur (After)], with respect to their own basal levels before taking curcumin [Anx-Cur (Before); r = −0.52, *p* = 0.012], according to the Spearman correlation analysis.

In addition, the Spearman correlation indicated a correlation between systemic sCD14 levels and anxiety scores before taking curcumin in patients with moderate anxiety [Anx-Cur (Before); r = 0.41, *p* = 0.05]. 

### 5.8. Spearman Correlations among Study Groups

The curcumin-supplemented group of patients with moderate anxiety presented a correlation between the Hamilton scores when Anx-Cur (After) was compared with their own basal levels before taking curcumin [Anx-Cur (Before); r = 0.68, *p* < 005, *n* = 22)]. In addition, there was a strong correlation between cortisol levels after taking curcumin [Anx-Cur (After)] and their own basal levels before taking curcumin [Anx-Cur (Before); r = 0.931, *p* = 0.0000002, *n* = 22].

### 5.9. Correlation between Cortisol and Biochemical Markers

Patients with moderate anxiety showed a positive correlation between Hamilton scores before/after taking curcumin (see Figure 5A; r = 0.68, *p* < 0.05). In addition, after 15 days of curcumin supplementation, we observed a negative correlation between cortisol and sCD14 levels in patients with moderate anxiety not taking curcumin (r = 0.47, *p* = 0.026, see Figure 5B); however, this correlation was positive for MCP-1 and sCD14 levels after 15 days of curcumin supplementation in the group of patients with moderate anxiety [Anx-Cur (After); r = 0.41, *p* = 0.05, *n* = 22; see Figure 5C].

The following graph illustrates all of the main findings in curcumin-treated patients, as a brief summary (see Figure 6).

## 6. Discussion

The awakening cortisol response in people with anxiety disorders can affect their anxiety levels [29,30,32,34]. As salivary cortisol levels (from 8 to 10 a.m.) are correlated with associated blood levels [27], we measured salivary cortisol levels as a good predictor of subclinical symptomatology after cortisol awakening in patients [29,30], including those with moderate anxiety. Our results are in line with the findings of Diotaiuti et al. (2021), who confirmed the importance of metric validity when analyzing interpersonal reactivity [13].

In our study, supplementation with curcumin (in powder form) was able to prevent the overproduction of cortisol associated with anxiety. In a recent study, the effects of nanocurcumin supplementation were studied in patients with cardiovascular disease and mental alterations, and their cardioprotective health improved after receiving this antioxidant. In fact, their scores were significantly better in the 36-item short-form quality of life survey (SF-36) as well as in Beck’s Depression Inventory-II (BDI-II) scale after nanocurcumin supplementation (80 mg/day) for 12 weeks [35]. Our findings agree with this clinical study, and the suggested anxiolytic effect of curcuminoids observed in our study is also supported by pre-clinical findings in rodent models of stress with *Curcuma longa* treatment [34,36]. The lower Hamilton score of curcumin-treated patients with moderate anxiety could reflect resilience against anxiety, in agreement with pre-clinical evidence in rodent models of stress and/or depression after curcumin supplementation [32]. These results are consistent with the better Hamilton scores observed in curcumin-treated patients with anxiety, which is also in line with the reported anxiolytic effects of nanocurcumin in other clinical studies when compared to placebo-treated subjects [18,19,35]. Thus, curcumin could reduce HPA overactivation by reducing cortisol levels, which is consistent with our results in patients with anxiety. In our study, the curcumin concentration used was 80 times higher (18,000 mg/kg) than in Sotani’s study [35]. Furthermore, curcuminoid treatment in the range of 500–1000 mg/day for 6–12 weeks has been reported in clinical studies for the treatment of major depression [36,37]. With regard to tolerability, there was an absence of toxicity in the curcumin-treated groups in our study, confirming the short-term safety profile of curcuminoids. In addition, short-term (<5 months) supplementation increased the bioavailability of curcuminoids without adverse effects in human studies [38,39], which is in agreement with our findings. Curcumin induced a variety of anti-inflammatory and antioxidant mechanisms in pre-clinical models through the activation of transcription factors (p65 NF-kappa beta and Nrf-2) and the induction of endogenous antioxidant enzyme systems as protective mechanisms [21].

The possible relationship between cortisol and peripheral MCP-1 overproduction remains to be investigated in patients with anxiety. In our study, curcumin prevented MCP-1 overproduction and decreased systemic sCD14 levels in patients with moderate anxiety. It is well known that stress enhances monocyte trafficking to the brain in mice with anxiety [14], and MCP-1 overproduction could contribute to the adverse effects of anxiety by inducing the peripheral mobilization of activated monocytes; in fact, the attenuated release of MCP-1 decreased the mobilization of monocytes in a rodent model of the disease [40], a phenomenon potentially reflected in our study findings. As curcumin reduces MCP-1 expression in vitro [18], our findings suggest that the decrease in MCP-1 levels could explain our results in curcumin-treated patients with anxiety.

We suggest that the overexpression of MCP-1 and sCD1 (a marker of monocyte activation) could be associated with anxiety-related behaviors through the induction of systemic monocyte activation. This feature is consistent with the monocyte mobilization induced by MCP-1 in rodent models [40,41,42]. Our findings in patients with anxiety also agree with the attenuated myeloperoxidase activity reported in pre-clinical models [42]; in addition, the MCP-1 blockade prevented monocyte recruitment in rodent models [33,40,41,42,43], suggesting that the chemotactic effects may have been dependent on the alteration in MCP-1 levels observed in our study. In fact, the neutralization of MCP-1 levels also abolished TNF-α overproduction in serum in LPS-treated mice [16,40], suggesting that inflammation can amplify monocyte mobilization in rodent models. In our study, curcumin maintained MCP-1 at normal levels and reduced anxiety-related behaviors, as reflected by reduced Hamilton scores. Maintaining MCP-1 levels in their normal range could be necessary for the resolution of inflammation in patients with anxiety, while excessive MCP-1 overproduction can accelerate the progression of anxiety-related behaviors, particularly considering that MCP-1 is also a marker of dementia [44].

To the best of our knowledge, no previous studies have analyzed the relationship between anxiety and systemic sCD14 levels in patients. The elevated sCD14 levels observed in patients with moderate anxiety suggest enhanced monocyte activation under anxiety. As peripheral inflammation increases monocyte mobilization into the neural route [11], we should not exclude the possibility that curcumin supplementation can reduce monocyte activation not only through decreasing systemic sCD14 levels but also by preventing MCP-1 overexpression as an anti-chemotactic effect in patients with moderate anxiety. Considering that MCP-1 directs the recruitment of monocytes under peripheral-induced inflammation in mice [33,40], the possibility that high MCP-1 levels could indirectly reflect HPA overactivation should not be excluded in patients with anxiety. The activation of monocytes by anxiety could activate systemic monocytes by increasing systemic sCD14 levels. As MCP-1 enhances the mobilization of monocytes in certain neuropsychiatric diseases [44,45] and elevated corticosterone levels have been observed in mice lacking MCP-1^−/−^ [40,46], we suggest that MCP-1 overexpression can contribute to anxiety-related behaviors associated with systemic sCD14 overproduction. However, curcumin supplementation did not affect MCP-1 or sCD14 levels among patients without anxiety, suggesting the relevance of anxiety in the mobilization of monocytes via sCD14 levels.

On the other hand, overexpression of the CCR2 chemokine receptor directed the recruitment of monocytes in rodent models when MCP-1 was bound to this chemokine receptor [47]. In fact, it is well known that the anxiety response enhances monocyte mobilization in rodent models of inflammation [48,49], indirectly supporting the involvement of monocytes in anxiety-related behaviors in patients. As such monocyte mobilization depends on CCR2 activation, we suggest that high systemic sCD14 and MCP-1 levels could reflect the mobilization of monocytes into the bloodstream of patients with anxiety. In fact, prolonged exposure to stress facilitates the passage of glucocorticoid-insensitive monocytes into circulation via CCR2 chemokine receptor activation (i.e., MCP-1 binds to CCR2) [47]. The suggested contribution of MCP-1 to anxiety is also supported by the abolished recruitment of monocytes into the brains of mice under CCR2 blockade [47,48]. The suggested contribution of activated monocytes to anxiety is also consistent with enhanced monocyte mobilization in rats with psychological distress due to sympathetic activation [50,51].

In our study, curcumin normalized systemic sCD14 and MCP-1 overproduction, which can be considered an anti-chemotactic response in patients with moderate anxiety, in agreement with indirect findings in pre-clinical models of inflammation [51,52]. These elevated systemic sCD14 levels detected in patients with anxiety were consistent with high systemic CD163 levels (another marker of monocyte activation) found in the bloodstream of COVID-19-infected patients, which could enhance monocyte–macrophage mobilization [53]. In fact, these elevated systemic CD163 levels were correlated with proinflammatory cytokine release (particularly IL-6) in COVID-19-infected patients. 

Our results also confirmed a negative correlation between sCD14 and MCP-1 levels under curcumin supplementation in patients with moderate anxiety. However, the opposite correlation was observed in sCD14/MCP-1 levels in anxiety patients not taking curcumin (before supplementation). These findings suggest that in patients with anxiety, patients with the highest cortisol levels, and patients with anxiety not taking curcumin, the systemic activation of monocytes is favored (see Figure 6). This evidence is supported by the re-establishment of anxiety in stress-sensitized mice caused by monocyte recruitment into the brain [52]. Collectively, curcuminoids from *Curcuma longa* not only promoted anxiolytic effects but also prevented systemic MCP-1 and sCD14 overexpression through curcumin supplementation in patients with moderate anxiety.

This pilot study cannot provide a conclusive and definite causal relationship between MCP-1 and sCD14 levels in patients with moderate anxiety. Further studies are required to evaluate the effects of long-term curcuminoid supplementation with larger sample sizes, as well as the influence of covariates, for a deeper understanding of our findings. We intend to evaluate different outcomes, including cardiometabolic and vascular factors and different families of systemic chemokines. Further studies will evaluate the roles of chemokines, p65 NF-kappa beta, and Nrf-2 in anxiety. The evaluation of these biochemical markers could serve to clarify the efficacy of long-term curcumin supplementation; finally, further studies should compare the clinical efficacy of curcumin in patients with different degrees of anxiety.

The Hamilton anxiety scale is based on patients’ subjective responses, which may not be valid when patients do not provide reliable answers. The integration of cortisol levels with additional behavioral tasks, such as the DASP-42 scale, could provide a better and more complementary assessment of anxiety status in patients. In addition, the psychological state of patients can be affected by non-medical factors (e.g., education state, socio-economic status).

In the future, we intend to evaluate the relevance of the Interpersonal Reactivity Index for predictive purposes in curcumin-treated patients, particularly for the assessment of empathy and individual psycho-educational interventions, including empathy as a motivating element of prosocial behavior in healthcare professionals [11]. Finally, the study of new nanocurcumin formulations is mandatory in order to increase their low bioavailability after oral administration. In fact, the means to address its poor water solubility and biological half-life are actively being investigated in pre-clinical models. Finally, the use of adjuvants (e.g., piperine) that can enhance the bioavailability and pharmacokinetic pitfalls of free curcumin is a very active research field.

## 7. Conclusions

This study makes a significant contribution to the field of psychology, suggesting that short-term curcumin supplementation for 15 consecutive days prevents systemic MCP-1 overproduction and decreases sCD14 levels in patients with moderate anxiety. In addition, curcumin induces anxiolytic effects by decreasing (salivary) cortisol levels and reducing Hamilton scale II scores. 

Future research will include the quantification of different families of chemokines in patients with different degrees of anxiety (alpha, delta, and so on).

## Figures and Tables

**Figure 1 antioxidants-13-01052-f001:**
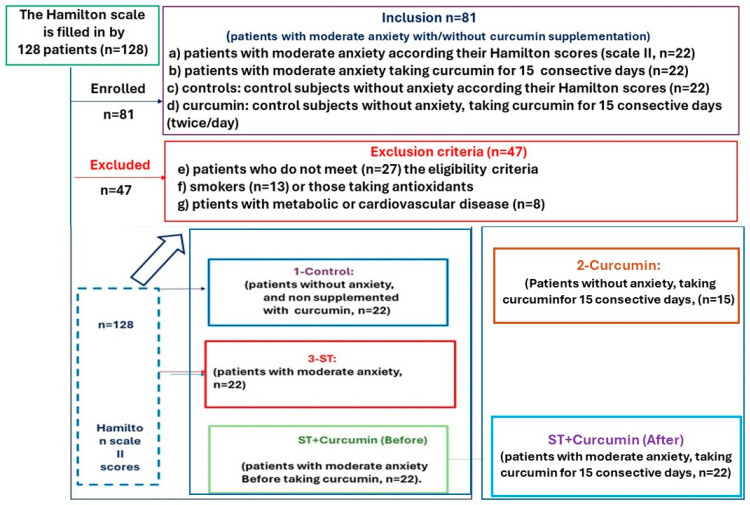
Scheme of enrolled participants and study groups.This scheme shows the enrollment of the 128 patients according to their initial Hamilton scores. This pilot study included 81 patients and 47 excluded subjects. Control (*n* = 15): subjects without anxiety (*n* = 22). Cur (*n* = 15): control patients without anxiety taking curcumin (powder phytosome form, Meriva@) for 15 consecutive days (1800 mg 2 times/day, *n* = 15). Anx (*n* = 22): patients with moderate anxiety (without taking curcumin, *n* = 22). The eligibility criteria for this group are scores within the range of moderate anxiety in the Hamilton scale II (Anx group, *n* = 22). Anx-Cur (After) (*n* = 22): patients with moderate anxiety taking curcumin for 15 consecutive days (1800 mg twice/day, *n* = 22). As an additional control, we compared systemic markers (pg/mL: MCP-1, sCD14, TNF alpha levels) in patients with anxiety after taking curcumin [Anx-Cur (After)] with their own basal levels [before taking curcumin, Anx-Cur (Before), *n* = 22]. Anx-Cur (Before) (*n* = 22): patients with moderate anxiety (before taking curcumin, *n* = 22).

**Figure 2 antioxidants-13-01052-f002:**
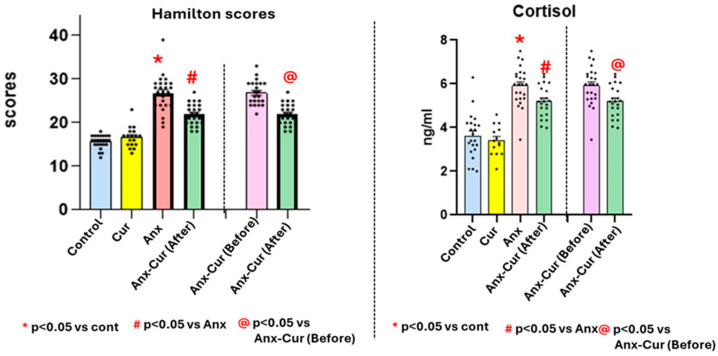
Hamilton scores and cortisol levels (ng/mL). Mean values ± S.E.M. for Hamilton scores (**left**) and salivary cortisol levels (ng/mL, right panel).

**Figure 3 antioxidants-13-01052-f003:**
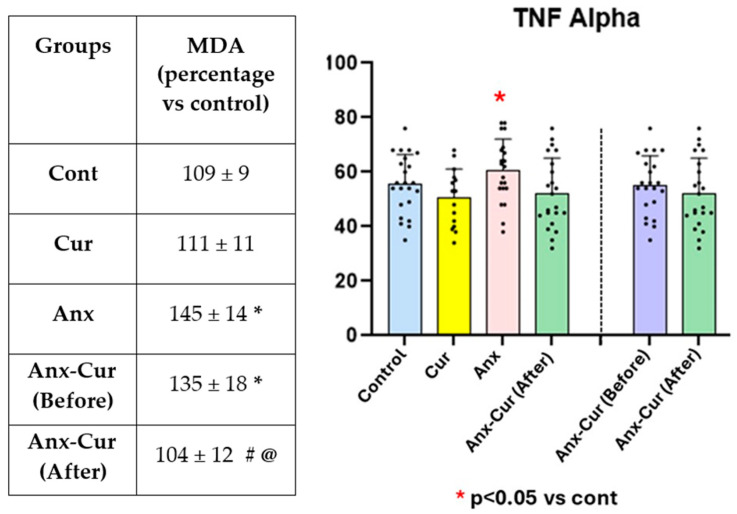
TBARS (MDA) indicating the percentage of MDA levels as compared to control (considered as 100%) (**left**) and mean values ± S.E.M. for systemic TNF alpha levels (pg/mL). * *p* < 0.05 vs. cont # *p* < 0.05 vs. Anx; # *p* < 0.05 vs. patients with moderate anxiety before curcumin treatment [Anx-Cur (Before)]; MDA percentages vs control (**left**) and mean values ± S.E.M. for systemic TNF alpha (pg/mL) levels (**right**). @ *p* < 0.05 vs Anx Cur (Before).

**Figure 4 antioxidants-13-01052-f004:**
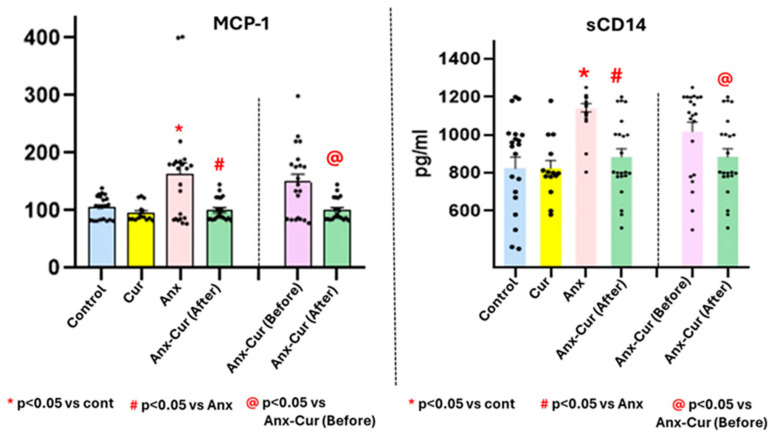
Mean values ± S.E.M. for systemic MCP-1 (pg/mL: left) and SCD14 levels (**right**: pg/mL). MCP-1 (pg/mL, **left**) and sCD14 (pg/mL) levels (right panel) levels via ELISA.

**Figure 5 antioxidants-13-01052-f005:**
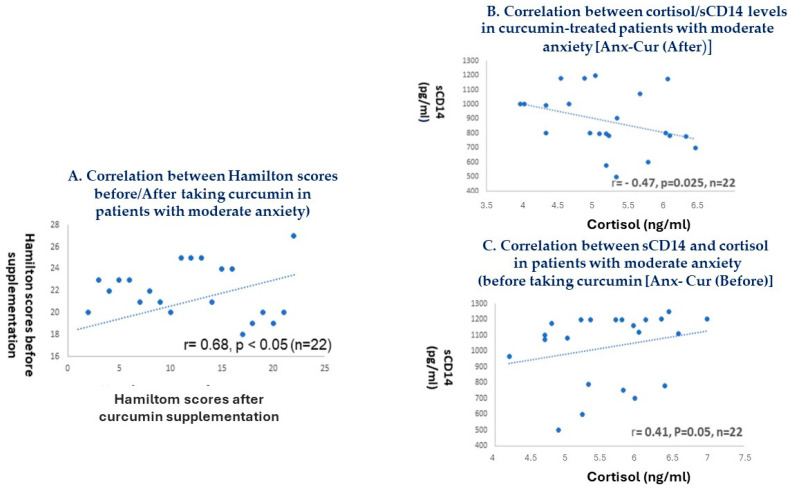
Spearman correlations between cortisol and systemic sCD14 protein levels in patients with moderate anxiety. Spearman correlations between Hamilton scores after/before taking curcumin (**A**); correlation between cortisol and sCD14 levels after 15 days of curcumin supplementation in patients with moderate anxiety [Anx-Cur (After)] (**B**); and correlation between cortisol and sCD14 levels in patients with moderate anxiety before taking curcumin [Anx-Cur (Before)] (**C**). Notably, there was a positive correlation between sCD14 and cortisol levels in patients with moderate anxiety before taking curcumin [Anx-Cur (Before)].

**Figure 6 antioxidants-13-01052-f006:**
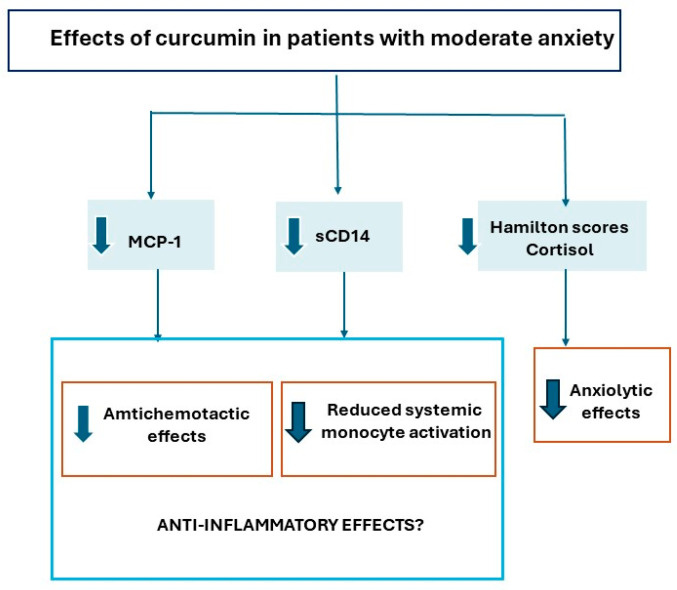
Summary of mean findings. Mean effects of curcumin supplementation in patients with moderate anxiety, mediated by decreased salivary cortisol levels and prevention of overexpression of systematic MCP-1 and sCD14 due to anxiety, suggesting anti-inflammatory effects.

**Table 1 antioxidants-13-01052-t001:** Clinical characteristics of enrolled patients.

Characteristics	Control (*n* = 22)	Cur (*n* = 15)	Anx (*n* = 22)	Anx + Cur (After) (*n* = 22)
BMI (Mean ± SEM)	21.5 ± 7	21 ± 2.73	23 ± 1.5	22.8 ± 3.1
Sex				
Female	16	11	14	17
Male	6	4	8	5
Age (years) Mean ± SEM	43 ± 7.9	45 ± 9	48 ± 8	46 ± 7.9
Sociocultural status	Medium/high	Medium/high	Medium/high	Medium/high

*Abbreviations: n*, number of patients; SEM: standard error of mean; BMI: body mass index; Lean (<25 kg/m^2^), overweight (25–29.9 kg/m^2^), and obese (≥30 kg/m^2^).

**Table 2 antioxidants-13-01052-t002:** Mean ± S.E.M. for Hamilton scores, salivary cortisol (ng/mL), and all systemic markers (pg/mL: MCP-1, sCD14, TNF alpha) quantified in study groups.

Marker (mean ± SEM)	Cont	Cur	Anx	Anx + Cur
Hamilton scores	15.59 ± 0.39	16 ± 0.44	27 ± 0.92 *	22 ± 0.5 * #
Cortisol (ng/mL)	3.2 ± 0.21	3.4 ± 0.18	6.45 ± 0.19 *	5.21 ± 0.15 * #
Malonaldehyde (MDA: % vs. control)	100 ± 9	101 ± 11	145 ± 14 *	104 ± 12 #
TNF Alpha	47 ± 1.6	50.3 ± 2.41	60 ± 1.6	52.3 ± 2.74 #
sCD14 (pg/mL)	843 ± 60	834.46 ± 41	1142 ± 22 *	883.9 ± 43 * #
MCP-1 (pg/mL)	104.9 ± 4.13	95 ± 3.94	173 ± 23 *	149.9 ± 13 * #

* *p* < 0.05 vs. cont. # *p* < 0.05 vs. patients with moderate anxiety (Anx). Cont: patients not taking curcumin (*n* = 22). Cur: patients taking curcumin for 15 consecutive days (*n* = 15, controls taking curcumin). Anx: patients with moderate anxiety not taking curcumin (*n* = 22). Anx-Cur (After): patients with moderate anxiety taking curcumin for 15 consecutive days (*n* = 22). We also compared these markers in patients with anxiety after taking curcumin with respect to their own basal values [before taking curcumin, ANX-Cur (Before), *n* = 22].

## Data Availability

All data are included within the manuscript. Any additional information required about data reported in this paper is available by contacting Dr. J. J. Merino (corresponding author).

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
