# Peer review of "Decreased Systemic Monocyte Colony Protein-1 (MCP-1) Levels and Reduced sCD14 Levels in Curcumin-Treated Patients with Moderate Anxiety: A Pilot Study"

_antioxidants, 2024, doi:10.3390/antiox13091052_

Round 1

Reviewer 1 Report

This manuscript by Merino et al examined the effect of curcumin on patients with anxiety. It shows that the stress-related amount of cortisol in saliva and monocyte’s activation are reduced by taking curcumin. The manuscript needs to be thoroughly revised before publication. It contains many typos. The figures need to be redesigned.

General comments 

1. No information is given about the origin of the curcumin. This means that the work is not reproducible. Where does the curcumin come from?

2. Curcumin has a very low bioavailability. It says that curcumin was given in powder form. Is this even bioavailable? Are there any studies on curcumin serum levels?

3. In the curcumin group with anxiety, cortisol and inflammatory markers are reduced. How can curcumin reduce the amount of cortisol? Please describe clearly in the discussion. What is the mechanism of curcumin? I also recommend a diagram for this. 4. Does curcumin treatment also alleviate the disease, the anxiety syndrome? 

Specific comments 

5. The manuscript is written sloppily. Here are just a few tips: Line 36: Psychosocial line 36: associated line 55: The study aimed to evaluate line 96: in powder form Line 109: the following ... 

Page 4: Here 5 groups were notified. Are the controls patients or normal individuals? I assume they were healthy subjects. Please clarify. 

Line 200 was isolated from peripheral ... Line 208: recent 

Page 11: The methods were described before. Why here again? Repetition. Please correct. 

Page 12: Table 2: Page 4 identifies 5 groups, in this table 4 groups were shown. Clarify. I recommend: 

Control 

Control with Cur 

Anx 

Anx before Cur 

Anx after Cur 

Page 13 and following: The figures must be numbered and combined into panels. They need sequential numbering and meaningful labeling.

Page 16: The figures here are not even explained and numbered.

Panels A and C. On page 17, panels B and D. – Quite confusing.

See my report

Author Response

Dear reviewer

Thanks for all your comments, which help us to improve this R1 version

We have improved the method section and also the introduction following your advice

The discussion has been shortented and the  number of references reduced from 73 to 53 following other reviewer advice.

Major comments

This manuscript by Merino et al examined the effect of curcumin on patients with anxiety. It shows that the stress-related amount of cortisol in saliva and monocyte’s activation are reduced by taking curcumin. The manuscript needs to be thoroughly revised before publication. It contains many typos. The figures need to be redesigned.

 The content has been improved following all suggested reviewer comments.

Following your advice. the figures were grouped and redesigned using Prisma 8.0

We also replace ST by Anx (Anxiety) since a reviewer other reviewer suggested us this modification in figures and text.

We have revised the manuscript. The english style has been check it by an english mother language expert. In addition, I have improved the introduction by including some mechanisms of curcuminods and its clinical efficacy such as required other reviewer.

Yes, curcumin is a food supplement. Wr used a phytosomed curcumin (Merive@ is the commertial name), which is a podwer form that meanly contains 95 % of curcuminoids (bisdemetoxicurcumin and demetoxicurcumine. This a commertial curcumin and standarized with controls of quality.

Thanks for your comments.

General comments 

No information is given about the origin of the curcumin. This means that the work is not reproducible. Where does the curcumin come from?

Yes, the work is reproducible. As a told you before, this is a commertial curcumin (Meriva@) that contain 95 % of standarized curcuminoids (bisdemetoxicurcumin and demetoxicurcumine) with a thecnology of phytosomes.

We administered 1800 mg/day of curcumin (oral intake) in all treated patients. Thus, the composition is standarized and this is a commertial curcumin with known composition. The name is curcumin Meriva®  (pytosomes curcumin, 95 % of purity).

Merive@ is a phtosomed powder curcumin by the combination of phospholipids with curcumin. This is a phytosomed formulation that contains curcuminoids with 95 % of purity (bisdemetoxi curcumin and demetoxicurcumin), which are detected in plasma and tissues (Marczylo TH, et al.. 2009). Merive@is a commertial phytosomed curcuminoids (Merive@) meanly has desmethoxycurcumin and bisdesmethoxycurcumin (Marczylo TH, et al. Rapid analysis of curcumin and curcumin metabolites in rat biomatrices using a novel ultraperformance liquid chromatography (UPLC) method. J Agric Food Chem. 2009 Feb 11;57(3):797-803). These findings in rats suggest that curcumin formulated with phosphatidylcholine furnishes higher systemic levels of parent agent than unformulated curcumin.

We have incorporated these references to the mathodology.

  1. Curcumin has a very low bioavailability. It says that curcumin was given in powder form. Is this even bioavailable? Are there any studies on curcumin serum levels?

Thanks for your comment.

Yes, curcumin (Meriva@) in powder form is bioavailable as phytosomed form. This phospholipid-phytochemical complexes improve gastrointestinal absorption of poorly water-soluble phytochemicals via the amphipathic properties of the phospholipid [Kidd, P.M. Bioavailability and activity of phytosome complexes from botanical polyphenols: The silymarin, curcumin, green tea, and grape seed extracts. Altern. Med. Rev. 2009, 14, 226–246].

The phospholipids formulation with lecithin increased the bioavailability of curcuminoids is similar to Merive@ formulation as phytosomed curcumin [Cuomo, J et al. Comparative absorption of a standardized curcuminoid mixture and its lecithin formulation. J. Nat. Prod. 2011, 74, 664–669.]. Other studies based on curcumin administration by lipids mixture or in complexation with phospholipids indicated only a mild increase in the bioavailability [Mati et al, 2006; Liu et al., 2006; Peng et al., 2018).

Maiti, K.; et al. Curcumin-phospholipid complex: Preparation, therapeutic evaluation and pharmacokinetic study in rats. Int. J. Pharm. 2007; Liu, A.; et al. Validated LC/MS/MS assay for curcumin and tetrahydrocurcumin in rat plasma and application to pharmacokinetic study of phospholipid complex of curcumin. J. Pharm. Biomed. Anal. 2006.

Peng, S.; et al. Enhancement of Curcumin Bioavailability by Encapsulation in Sophorolipid-Coated Nanoparticles: An in vitro and in vivo Study. J. Agric. Food Chem. 2018, 66, 1488–1497

These studies clearly indicate that phytosomed curcumin by encapsulation with phospholipid as the present used curcumin in our study increase the availability of curcuminoids. Thus, the used Meriva@ curcumin is this study has good availability, also in patients.

In rats, peak plasma concentration and AUC were 5-fold higher for Meriva (a combination of curcumin-phospholipids) than for unbound curcumin [Marczylo, et al. Comparison of systemic availability of curcumin with that of curcumin formulated with phosphatidylcholine. Cancer Chemother. Pharmacol. 2007]. Another small single-dose study demonstrated a comparable absorption of curcumin from 450 mg of Meriva and 4 g unbound of Curcuma longa [Jurenka, J.S. Anti-in ammatory Properties of Curcumin, a Major Constituent of Curcuma longa: A Review of Preclinical and Clinical Research. Altern. Med. Rev. 2009, 14, 141–153]. The concentration of curcumin used in our study was 1800 mg/day during 15 consecutive days. In a study, plasma and rectal tissue concentrations of curcuminoid were studied between standard curcumin and phosphatidylcholine-complexed curcumin preparations as Merive@. Plasma concentrations of curcumin and the major curcuminoids in curcumin extracts were similar even though the doses were 4 g for pure curcumin and 400 mg for phosphatidylcholine-complexed curcumin [Asher, G.N.; et al. Randomized Pharmacokinetic Crossover Study Comparing 2 Curcumin Preparations in Plasma and Rectal Tissue of Healthy Human Volunteers. J. Clin. Pharmacol. 2017, 57, 185–193]. In my opinion, I thing 4 g is a very high dose for humans although no toxic effects have been observed at high doses of curcuminoids.

The availability of curcumin has been also enhanced by combination with piperine in other kind of formulations. In fact, 8 healthy male volunteers after 2 g/kg of oral pure curcumin intake showed very low serum levels of curcumin (Cmax 0.006 ± 0.005 µg/mL at 1 h), however, when 2 g/kg of pure curcumin are combined with 20 mg/kg of piperine the concentrations are significantly increased (0.18 ± 0.16 µg/mL at 0.75 h) [Shobal, G.; et al. Influence of Piperine on the Pharmacokinetics of Curcumin in Animals and Human Volunteers. Panta Medica 2000, 64, 353–356.].

In rats, peak plasma concentration and AUC were 5-fold higher for Meriva (a combination of curcumin-phospholipids) than for unbound curcumin [Marczylo, T.H.; et al. Comparison of systemic availability of curcumin with that of curcumin formulated with phosphatidylcholine. Cancer Chemother. Pharmacol. 2007]. Another small single-dose study demonstrated a comparable absorption of curcumin from 450 mg of Meriva and 4 g unbound of Curcuma longa [58]. In a study, plasma and rectal tissue concentrations of curcuminoid were studied between standard curcumin and phosphatidylcholine-complexed curcumin preparations. Plasma concentrations of curcumin and the major curcuminoids in curcumin extracts were similar even though the doses were 4 g for pure curcumin and 400 mg for phosphatidylcholine-complexed curcumin [Asher, G.N.; et al.. Randomized Pharmacokinetic Crossover Study Comparing 2 Curcumin Preparations in Plasma and Rectal Tissue of Healthy Human Volunteers. J. Clin. Pharmacol. 2017, 57, 185–193].

A study in rats have evaluated the availability of curcuminoids and its metabolites after oral intake in rats. In this study, the systemic availability of curcumin and curcumin formulated with phosphatidylcholine (Merive@) were compared in 340 mg/Kg curcumin-treated rats. In fact, the poor systemic availability of curcumin is enhanced by the formulation with phosphatidylcholine in Merive@; the results confirmed increased oral bioavailability in phytosomed-curcumin (Merive@) treated rats and also Merive@ affects the metabolite profile of curcumin in rats (Marczylo TH, et al. Rapid analysis of curcumin and curcumin metabolites in rat biomatrices using a novel ultraperformance liquid chromatography (UPLC) method. J Agric Food Chem. 2009 Feb 11;57(3):797-803)

In this study, male Wistar rats received 340 mg/kg of either unformulated curcumin or curcumin formulated with phosphatidylcholine (Meriva) by oral gavage. Rats were killed at 15, 30, 60 and 120 min post administration. The presence of curcumin was evaluated in plasma, intestinal mucosa and liver were analysed for the presence of curcumin and metabolites using HPLC with UV detection. This study identidy curcumin, the accompanying curcuminoids desmethoxycurcumin and bisdesmethoxycurcumin, and the metabolites tetrahydrocurcumin, hexahydrocurcumin, curcumin glucuronide and curcumin sulfate in plasma, intestinal mucosa and liver of rats which had received Meriva. Peak plasma levels and area under the plasma concentration time curve (AUC) values for parent curcumin after administration of Meriva were fivefold higher than the equivalent values seen after unformulated curcumin. Similarly, liver levels of curcumin were higher after administration of Meriva as compared to unformulated curcumin. In contrast, curcumin concentrations in the gastrointestinal mucosa after ingestion of Meriva were somewhat lower as compare to other curcumin. Similar observations were made for curcumin metabolites as for parent compound.

  1. In the curcumin group with anxiety, cortisol and inflammatory markers are reduced. How can curcumin reduce the amount of cortisol? Please describe clearly in the discussion. What is the mechanism of curcumin? I also recommend a diagram for this. 4. Does curcumin treatment also alleviate the disease, the anxiety syndrome? 

Thanks for your important comment. We have modifed the discussion, including evidences about the anxiolotic role of curcuminoids and our observed antininflammatory and/or antioxidante effects by reducing lipoperoxidation in patients with moderate anxiety. We have added a diagram with the general antioxidant and/or antiinflammatory effects of curcuminoids in patients with anxiety.

How can curcumin reduce the amount of cortisol? Please describe clearly in the discussion

In fact, phytosomed curcumin plays antichemotactic effects by reducing MCP-1 levels and decrease systemic sCD14 levels. This sCD14 reduction could prevent monocyte mobilization into the bloodstream of patients with moderate anxiety after taking curcumin. We added these points to the discussion, including more evidences about the anxiolitic effects of curcuminoids in our study.

 I also recommend a diagram for this.

We have included a graphical abstract following your advice, which include a brief graph with our mean findings in curcumin-treated patients.

Does curcumin treatment also alleviate the disease, the anxiety syndrome? 

Yes, it you revised the Hamilton graph, the scores showed a signifficant effecf by decreasingd in scores after curcumin treatment in patients with anxiety as compare to those patients with moderate anxiety (without curcumin treatment). In fact, more that 60 % of patients with moderate anxiety decreased their Hamilton scores after 15 days of supplementation. Thus, 15 days of curcumin supplementation reduces anxiety (in general) although exist individual differences in patients with anxiety.

Other antiinflammatory mechanism of curcuminoids that we have not evaluated is the induced p65-Nf kappa beta nuclear transactivation by curcumin, leading to antiinflammatory effects by decreasing IL-6 levels; in addition, several studies have demonstrated antioxidant properties of curcuminoids by inducing Nrf-2 transcription factor. In general, curcuminoids prevents microglia overactivation and reduces astrogliosis as neuroprotection mechanisms in rodent models of neurological diseases.

The follows evidences showing antioxidant effects of curcumin are indicated here but are not included in the discussion because are under the scope of our study but will be evaluated in the future. These preclinical and clinical findings also confirm the safety of phytosome curcumin form after long-term administration, even in patients with different pathologies.

At the systemic level, the antioxidant effects of curcumin are evident on human individuals. In a randomized, parallel-group, placebo-controlled study, curcumin plays antioxidant effects in patients with diabetes mellitus after curcumin supplementation [Usharani P et al Effect of NCB-02, Atorvastatin and Placebo on Endothelial Function, Oxidative Stress and Inflammatory Markers in Patients with Type 2 Diabetes Mellitus: A Randomized, Parallel-Group, Placebo-Controlled, 8-Week Study. Drugs R D. 2008, 9:243–250]. The antioxidant effect of curcumin was seen also in a single-blind, randomized study where 20 patients with tropical pancreatitis showed a significant reduction in the erythrocyte MDA levels after curcumin treatment [Durgaprasad S., et al. A pilot study of the antioxidant effect of curcumin in tropical pancreatitis. Indian J. Med. Res. 200, ;122:315–318].

The anti-cytokine effect of curcumin phytosome (Meriva® ) has been demonstrated in different pathological conditions. In particular, curcumin phytosome supplementation induced PPARγ and reduced NFkB (p65 nuclear transactivaton) in a transgenic mouse model of hepatitis B virus-related hepatocellular carcinoma. PPARγ regulate a large number of genes involved in anti-inflammatory effects [Teng C.F., et al. Chemopreventive Effect of Phytosomal Curcumin on Hepatitis B Virus-Related Hepatocellular Carcinoma in A Transgenic Mouse Model. Sci. Rep. 2019;9:1–13. doi: 10.1038/s41598-019-46891-5]. Moreover, this used Meriva@ formulation with curcumin can prevent astrogliosis and decreased IL6 levels in a rodent model of neuroinflammation, demonstrating that phytosomal curcumin is able to attenuate the inflammatory pathology [Ullah F., et al. Evaluation of Phytosomal Curcumin as an Anti-inflammatory Agent for Chronic Glial Activation in the GFAP-IL6 Mouse Model. Front. Neurosci. 2020, ;14:170.]. An interesting randomized controlled trial on forty COVID-19 patients supplemented with nano-curcumin showed a significant reduction in IL-1β and IL-6 expression and secretion [Valizadeh H.., et al. Nano-curcumin therapy, a promising method in modulating inflammatory cytokines in COVID-19 patients. Int. Immunopharmacol. 2020;89:107088]. The same pro-inflammatory cytokines reduction was demonstrated in patients with osteoarthritis, in supplemented patients with 1 g daily of phytosomal curcumin for 8 months [Belcaro G., et al. Efficacy and safety of Meriva®, a curcumin-phosphatidylcholine complex, during extended administration in osteoarthritis patients. Altern Med. Rev. 2010;15:337–344]. These published studied confirmed the antiinflammatory and protective effects of phytosomed curcumine in rodent models of diseases as well in treated patients with certain pathologies.

Several preclinical studies reported increased antioxidant enzymatic activity by curcumin supplementation [Lin X., et al.Curcumin attenuates oxidative stress in RAW264.7 cells by increasing the activity of antioxidant enzymes and activating the Nrf2-Keap1 pathway. PLoS ONE. 2019;14:e0216711;  Samarghandian S., et al. Anti-oxidative effects of curcumin on immobilization-induced oxidative stress in rat brain, liver and kidney. Biomed. Pharmacother. 2017;87:223–229].

In fact, in vitro studies have demonstrated several antioxidant effects of curcuminoids by reducing circulating ROS and lipid peroxidation phenomena by reducing conjugated diene and lipid peroxides in oxidized LDL isolated from human plasma [Mahfouz M.M., et al. Curcumin prevents the oxidation and lipid modification of LDL and its inhibition of prostacyclin generation by endothelial cells in culture. Prostaglandins Other Lipid Mediat. 2009;90:13–20]. Sadeghi A. and colleagues described the role of curcumin in ameliorating the inflammatory responses stimulated by palmitate in muscular cells by represing the phosphorylation of IKKα, IKKβ, and JNK and also decreased ROS levels [Sadeghi A., et al. Curcumin ameliorates palmitate-induced inflammation in skeletal muscle cells by regulating JNK/NF-kB pathway and ROS production. Inflammopharmacology. 2018;26:1265–1272].

Specific comments 

  1. The manuscript is written sloppily. Here are just a few tips: Line 36: Psychosocial line 36: associated line 55: The study aimed to evaluate line 96: in powder form Line 109: the following

We have included these suggested modifications in the R1 version. Please, take into account the number of pages and lines are different after adding all required modifications by reviewers in this R1 version.

Thanks for your comments. We have corrected these errors.

Page 4: Here 5 groups were notified. Are the controls patients or normal individuals? I assume they were healthy subjects. Please clarify. 

Yes, controls are healthy participants without anxiety. In addition, they are not supplemented with this curcumin. These control subjects are healthy patients without any pathological condition.

Line 200 was isolated from peripheral ... Line 208: recent 

Page 11: The methods were described before. Why here again? Repetition. Please correct. 

Please, take into account these lines are different now in this R1 version. Anyway, we included your comments also.

We have deleted since is repeated twice. Thanks¡. Again, the number of pages has changed in this R1 version.

Page 12: Table 2: Page 4 identifies 5 groups, in this table 4 groups were shown. Clarify. I recommend: 

Control 

Control with Cur 

Anx 

Anx before Cur 

Anx after Cur 

All figures were redone by using Prisma 8.0. I understand the created figures with Sigma Plot 11.0 program in the original submission are more difficult to follow. Thus, we replace by these color figures with all values +- S.E.M (Standard error media) that showed individual values for each study group.

We thing is better to put Cur instead of Control with Cur to avoid confusion with untreated controls (Cont). The rest of modifications have been done.

Page 13 and following: The figures must be numbered and combined into panels. They need sequential numbering and meaningful labeling.

Done it. Thanks¡

Page 16: The figures here are not even explained and numbered.

Panels A and C. On page 17, panels B and D. – Quite confusing.

We explained these figures and also at the end of discussion was added the clinical relevance of these correlations now in this R1 version.

Following your advice, we have replaced ST by Anx (moderate anxiety) in all the tables, text, including figures. These figures AB, C has been group in a figure.

We have explained the content by including foots of figures. Please, notice we have combined figures with two markers in order to reduce the number of singles figures.

Thanks again for all your comments.

Reviewer 2 Report

Thank you for giving me the opportunity to review the manuscript. I have identified several strengths, but there are also several areas that need revision to make it publishable. Below, I will provide a detailed analysis of the necessary changes, organized by page and line.

Strengths

  1. Relevance of the topic: The research on the effects of curcumin on anxiety is current and of scientific interest.
  2. Study design: The randomized clinical study design is well-conceived, and the number of participants is adequate.
  3. Significant results: The effects of curcumin on reducing cortisol and inflammatory markers are promising.

Necessary Modifications

1. Abstract (Page 1, Lines 13-31)

  • Modification needed: Correct grammatical errors and improve the clarity of the text.
  • Example of modification: Change "We have evaluated whether curcumin supplementation during 15 consecutive days (1800 mg/day) may decrease systemic MCP-1, sCD14 and TNF Alpha levels in patients with moderate anxiety (n=81)." to "We evaluated whether curcumin supplementation for 15 consecutive days (1800 mg/day) could reduce systemic MCP-1, sCD14, and TNF Alpha levels in patients with moderate anxiety (n=81)."

2. Introduction (Page 1, Lines 35-44)

  • Modification needed: Improve the flow and coherence of the text.
  • Example of modification: Change "Pychosocial stress in patients has been associate to high Cortisol levels in saliva and plasm and can contribute to detrimental consequences of chronic stress..." to "Psychosocial stress in patients has been associated with high cortisol levels in saliva and plasma, and can contribute to detrimental consequences of chronic stress..."

3. Methods (Page 3, Lines 61-88)

  • Modification needed: Correct spelling and grammatical errors and improve the clarity of the text.
  • Example of modification: Correct "This clinical randomized study enrrolled a total amount of 81 patients" to "This randomized clinical study enrolled a total of 81 patients."

4. Results (Page 12, Lines 295-301)

  • Modification needed: Strengthen the interpretation of the results, highlighting statistical relevance.
  • Example of modification: Include phrases such as "The results showed a significant reduction in cortisol levels in curcumin-treated patients, with a p-value of less than 0.05, indicating a statistically significant effect."

5. Discussion (Page 20, Lines 376-429)

  • Modification needed: Strengthen the connection between the results and the existing literature.
  • Example of modification: Integrate the discussion with references to other studies that support the findings, for example, "These results are consistent with previous studies where curcumin has shown anti-inflammatory effects in preclinical models of anxiety."

6. Conclusions (Page 23, Lines 511-513)

  • Modification needed: Improve the synthesis of the conclusions.
  • Example of modification: Change "The short-term supplementation with curcumin during 15 consecutive days in patients with moderate anxiety prevented systemic MCP-1 and reduced sCD14 levels..." to "Short-term curcumin supplementation for 15 consecutive days in patients with moderate anxiety prevented systemic MCP-1 elevation and reduced sCD14 levels..."

General Suggestions

  • Stylistic uniformity: Ensure that the scientific language is uniform throughout the manuscript.
  • Bibliographical references: Verify the accuracy and formatting of the bibliographical references.
  • Final revisions: After making the changes, a final read-through is recommended to check the manuscript's coherence.

I recommend including the following citation in your manuscript to enrich the theoretical context and provide a significant reference to existing literature:

Diotaiuti P, Falese L, Mancone S, and Purromuto F (2017) A Structural Model of Self-Efficacy in Handball Referees. Front. Psychol. 8:811. doi: 10.3389/fpsyg.2017.00811

This article could be particularly useful for discussions on self-efficacy, behavior under stress, or other relevant psychological aspects.

Thank you for giving me the opportunity to review the manuscript. I have identified several strengths, but there are also several areas that need revision to make it publishable. Below, I will provide a detailed analysis of the necessary changes, organized by page and line.

Strengths

  1. Relevance of the topic: The research on the effects of curcumin on anxiety is current and of scientific interest.
  2. Study design: The randomized clinical study design is well-conceived, and the number of participants is adequate.
  3. Significant results: The effects of curcumin on reducing cortisol and inflammatory markers are promising.

Necessary Modifications

1. Abstract (Page 1, Lines 13-31)

  • Modification needed: Correct grammatical errors and improve the clarity of the text.
  • Example of modification: Change "We have evaluated whether curcumin supplementation during 15 consecutive days (1800 mg/day) may decrease systemic MCP-1, sCD14 and TNF Alpha levels in patients with moderate anxiety (n=81)." to "We evaluated whether curcumin supplementation for 15 consecutive days (1800 mg/day) could reduce systemic MCP-1, sCD14, and TNF Alpha levels in patients with moderate anxiety (n=81)."

2. Introduction (Page 1, Lines 35-44)

  • Modification needed: Improve the flow and coherence of the text.
  • Example of modification: Change "Pychosocial stress in patients has been associate to high Cortisol levels in saliva and plasm and can contribute to detrimental consequences of chronic stress..." to "Psychosocial stress in patients has been associated with high cortisol levels in saliva and plasma, and can contribute to detrimental consequences of chronic stress..."

3. Methods (Page 3, Lines 61-88)

  • Modification needed: Correct spelling and grammatical errors and improve the clarity of the text.
  • Example of modification: Correct "This clinical randomized study enrrolled a total amount of 81 patients" to "This randomized clinical study enrolled a total of 81 patients."

4. Results (Page 12, Lines 295-301)

  • Modification needed: Strengthen the interpretation of the results, highlighting statistical relevance.
  • Example of modification: Include phrases such as "The results showed a significant reduction in cortisol levels in curcumin-treated patients, with a p-value of less than 0.05, indicating a statistically significant effect."

5. Discussion (Page 20, Lines 376-429)

  • Modification needed: Strengthen the connection between the results and the existing literature.
  • Example of modification: Integrate the discussion with references to other studies that support the findings, for example, "These results are consistent with previous studies where curcumin has shown anti-inflammatory effects in preclinical models of anxiety."

6. Conclusions (Page 23, Lines 511-513)

  • Modification needed: Improve the synthesis of the conclusions.
  • Example of modification: Change "The short-term supplementation with curcumin during 15 consecutive days in patients with moderate anxiety prevented systemic MCP-1 and reduced sCD14 levels..." to "Short-term curcumin supplementation for 15 consecutive days in patients with moderate anxiety prevented systemic MCP-1 elevation and reduced sCD14 levels..."

General Suggestions

  • Stylistic uniformity: Ensure that the scientific language is uniform throughout the manuscript.
  • Bibliographical references: Verify the accuracy and formatting of the bibliographical references.
  • Final revisions: After making the changes, a final read-through is recommended to check the manuscript's coherence.
  • I recommend including the following citation in your manuscript to enrich the theoretical context and provide a significant reference to existing literature:

    Diotaiuti P, Falese L, Mancone S, and Purromuto F (2017) A Structural Model of Self-Efficacy in Handball Referees. Front. Psychol. 8:811. doi: 10.3389/fpsyg.2017.00811

    This article could be particularly useful for discussions on self-efficacy, behavior under stress, or other relevant psychological aspects.

Author Response

Reviewer-2

Major comments

Thank you for giving me the opportunity to review the manuscript. I have identified several strengths, but there are also several areas that need revision to make it publishable. Below, I will provide a detailed analysis of the necessary changes, organized by page and line.

Strengths

  1. Relevance of the topic: The research on the effects of curcumin on anxiety is current and of scientific interest.
  2. Study design: The randomized clinical study design is well-conceived, and the number of participants is adequate.
  3. Significant results: The effects of curcumin on reducing cortisol and inflammatory markers are promising.

Necessary Modifications

Dear Reviewer

Thanks for all your comments, which help us to improve the manuscript content.

  1. Abstract (Page 1, Lines 13-31)
  • Modification needed: Correct grammatical errors and improve the clarity of the text.

Done it.

The english style has been revised by a native english mother expert.

Thanks¡

  • Example of modification: Change "We have evaluated whether curcumin supplementation during 15 consecutive days (1800 mg/day) may decrease systemic MCP-1, sCD14 and TNF Alpha levels in patients with moderate anxiety (n=81)." to "We evaluated whether curcumin supplementation for 15 consecutive days (1800 mg/day) could reduce systemic MCP-1, sCD14, and TNF Alpha levels in patients with moderate anxiety (n=81)."

Thanks¡. This modifcation has been introduced in this R1 version.

In addiiton, we have added all reviewer,s suggestions ìn this R1 version.

  1. Introduction (Page 1, Lines 35-44)
  • Modification needed: Improve the flow and coherence of the text.

Done it¡. Thanks¡

  • Example of modification: Change "Pychosocial stress in patients has been associate to high Cortisol levels in saliva and plasm and can contribute to detrimental consequences of chronic stress..." to "Psychosocial stress in patients has been associated with high cortisol levels in saliva and plasma, and can contribute to detrimental consequences of chronic stress..."

Please, take into account the number of pages and lines are different after adding all reviewer`s comments in this R1 version

This change has been introduced. Thanks¡

These lines have been redone it in page 1. Please, also consider the introduction has been redone following advice of other reviewer.

  1. Methods (Page 3, Lines 61-88)
  • Modification needed: Correct spelling and grammatical errors and improve the clarity of the text.
    Done it. Thanks¡

The manuscript has been also revised by an english mother language.

Please, take into account the introduction has been extended as I told you before.

  • Example of modification: Correct "This clinical randomized study enrrolled a total amount of 81 patients" to "This randomized clinical study enrolled a total of 81 patients."

Done it.

  1. Results (Page 12, Lines 295-301)
  • Modification needed: Strengthen the interpretation of the results, highlighting statistical relevance.

We have corrected the text and added the  follows sentence and others in this R1 version.

The pos Hoc analysis showed that patient with moderate anxiety had highscores tan controls. Interestingly, Hamilton scores were reduced after 15 days of curcumin supplementation as compare their own basal values (before taking curcumin, Anx-Cur (Before; p<0.05 in all tested cases).

  • Example of modification: Include phrases such as "The results showed a significant reduction in cortisol levels in curcumin-treated patients, with a p-value of less than 0.05, indicating a statistically significant effect."

Done it¡. We also added this extra setence for cortisol in this R1 version and part of the text are different (in text content) after following reviewe,s comments.

The post hoc analysis confirmed significant cortisol decreased levels in curcumin-treated patients as compare to controls (without anxiety), with a p-value of less than 0.05, indicating a statistically significant effect. In addition, a significant reduction in cortisol levels were observe after 15 days of curcumin supplementation in patients with anxiety (Anx-Cur (After) as compare their own basal values (before taking curcumine, Anx-Cur (Before), with a statistically significant effect.

These sentences were added to the methods following your advice.

  1. Discussion (Page 20, Lines 376-429)
  • Modification needed: Strengthen the connection between the results and the existing literature.

Done it. The discusión connect findings about anxiolytic effects of curcuminoids in the literature with our findings. In addition, the discussion has been shortened and also added limitations follwing reviwer advice in this R1 version. This is the reason by which lmitations are present now. We also regrouped the figures, including individual data of results with the Prisma 8.0 software. We understant are difficult to follow with the Sigma Plot 11.0 software in the original submission.

  • Example of modification: Integrate the discussion with references to other studies that support the findings, for example, "These results are consistent with previous studies where curcumin has shown anti-inflammatory effects in preclinical models of anxiety."

Done it¡. This sentence has been included within the discussion by integrating preclinical results and evidences in curcumin-treated subjects.

  1. Conclusions (Page 23, Lines 511-513)
  • Modification needed: Improve the synthesis of the conclusions.

The conclussion was improved by including relevant information. Thanks¡

  • Example of modification: Change "The short-term supplementation with curcumin during 15 consecutive days in patients with moderate anxiety prevented systemic MCP-1 and reduced sCD14 levels..." to "Short-term curcumin supplementation for 15 consecutive days in patients with moderate anxiety prevented systemic MCP-1 elevation and reduced sCD14 levels..."

Done it.

General Suggestions

  • Stylistic uniformity: Ensure that the scientific language is uniform throughout the manuscript.

The text content has been uniformed and evised by a english mother language expert.

  • Bibliographical references: Verify the accuracy and formatting of the bibliographical references.

The bibliography has been formated accoding to the Journal. This time the number of references is 53 (the original submission had 73).

  • Final revisions: After making the changes, a final read-through is recommended to check the manuscript's coherence.

Yes, the manuscript has been revised by an english native expert in Science.

I recommend including the following citation in your manuscript to enrich the theoretical context and provide a significant reference to existing literature:

Diotaiuti P, Falese L, Mancone S, and Purromuto F (2017) A Structural Model of Self-Efficacy in Handball Referees. Front. Psychol. 8:811. doi: 10.3389/fpsyg.2017.00811

This reference has been included in the introduction. Thanks¡

The number of cyte is 13.

This article could be particularly useful for discussions on self-efficacy, behavior under stress, or other relevant psychological aspects.

Yes, we have readed and its interesting, spetially for statistical interpretation and behavior. Thanks again¡

Detail comments

Thank you for giving me the opportunity to review the manuscript. I have identified several strengths, but there are also several areas that need revision to make it publishable. Below, I will provide a detailed analysis of the necessary changes, organized by page and line.

The follows part appaers appear twice repeated.

So, I understand it is not neccesary reply our response twice.

Thanks for all your comments again. 

Reviewer 3 Report

This is a potentially interesting study but I have a number of difficulties with the study design and presentation of the findings:

First, we are told all the subjects were patients but it is not clear what conditions the non-anxious controls had and whether they were being treated. It also isn’t stated whether patients in the two anxiety cohorts were assigned randomly to have curcumin or to have no treatment. Were the anxiety cases blinded as to whether they were being given curcumin or a placebo?

Second, , the multiple significant findings presented here are nearly all at a p<.05 threshold. Given the large number of comparisons and correlations this threshold will not have protected agaisnst false positives.

Third, the paper is stated to be about the efficacy and mechanism of action of curcumin in anxiety but the discussion focusses mainly on the possible functions of MCP-1 based on evidence from LPS rodent models of inflammation.

The English text is full of typos, grammatical errors, and mistakes  - in the abstract the curcumin supplements were said to be 1800 mg and later 18000 mg while in Fig 1A the ST and ST+CUR results for MCP-1 are the wrong way around. As it stands the report is difficult to read and is over long with an unfocussed discussion that generates a bibliography of 70 references - more typical of a review.

Author Response

Reviewer-3

Comments for Authors

Advice for completing your review can be found at: https://www.mdpi.com/reviewers#Review_Report

Major comments

This is a potentially interesting study but I have a number of difficulties with the study design and presentation of the findings:

First, we are told all the subjects were patients but it is not clear what conditions the non-anxious controls had and whether they were being treated. It also isn’t stated whether patients in the two anxiety cohorts were assigned randomly to have curcumin or to have no treatment. Were the anxiety cases blinded as to whether they were being given curcumin or a placebo?

Dear reviewer

Thanks for all your comments, which allow us to improve the content in this R1 version.

The study design includes the enrolment of all subjects (n=81) (see figure-1 in this R1 version). The allocation and treatments are also included in material and methods and we also added a chart flow at the end of discussion as supplementary material.

Thanks again¡

The enrolment of subjects with moderate anxiety, who were randomized (1:1) to receive the pytosomed curcumin (Meriva@) or placebo (control subjects without anxiety). 128 subjects were evaluated for their Hamilon scores and we enrolled 81 according their reached Hamilson scores; 44 participants have moderate anxiety and 47 subjects decline participate. Subjects were subsequently randomized to receive the nutraceutical (curcumin) or placebo for 15 conseutive days. At the end of cucrumin supplementation (day 15), they also fill up the Hamilton scores again.

This randomized pilot study enrolled a total of 81 patients. 22 of them were enrolled according their scores in the Hamilton scale II by testing several items, which belong to degree of moderate anxiety. Initially, 128 subjects fill up the Hamilton score  and they were assesed into four different stdy groups their reached scores; thus, patients were elegibility if they have moderate anxiety scores in the Hamilton scale II; we enrolled 44 patients with moderate anxiety, half of them (n=22) received placebo [n=22, Anxiety group (Anx), without taking curcumin], and 22 patients taking curcumin during 15 consecutive days [(Anx-Cur (After) group: 1800 mg/day,twice,  n=22]. The control are subjects without anxiety (Control, n=22) and also control subject without taking curcumin (Cur, n=15). All patients were aleatoreally assement in study groups according their Hamilton scores (see the enrolment in figure-1).

All supplemented patients received commertial curcumin in powder phytosomed form with phosphatydilserine (Merive@) during 15 consecutive days (twice, 1800 mg/day, from 8.00-10.00 A.M). Merive@ curcumin has 95 % of purity of curcuminoids (desmethoxycurcumin and bisdesmethoxycurcumin), which are detected in plasma and tissues [22].

As additional control, we compared all evaluated biomarkers in curcumin-treated patients with moderate anxiety after 15 days of curcumin supplementation [Anx-Cur (After) group, n=22, 1800 mg/day, twice/day), as compare their own basal levels [before taking curcumin: Anx-Cur (Before) group, n=22)]. Hamilton scores, cortisol and several systemic proinflammatory markers (TNF Alpha, MCP-1), sCD14 were compared between study groups. Hamilton and salivary cortisol levels (ng/ml) [26,27] and blood samples were collected during the first day (day 1: visit to the clinic as well 15 days after curcumin supplementation in subjects (day 15).

All obtained results were blinded physicians and researchers until the end of the study (see figure-1).

Please, notice that change the word Stress by Anxiety (Anx in th graphs and text), following reviewer advice.

Control: control subjects without anxiety (n=22).

Control Cur (n=15): patients without anxiety, taking curcumin (powder phytosomed form) during 15 consecutive days (1800 mg/day, 2 times/day).

Anx: patients with moderate anxiety (without taking curcumin supplementation).

Anx-Cur (After): patients that have with moderate anxiety according their Hamilton scores, taking curcumin during 15 consecutive days (1800 mg/day, 2 times/day, n=22).

In addition, all quantified systemic markers (pg/ml: MCP-1, sCD14, TNF alpha levels) were compared after taking curcumin [Anx-Cur (After)] and their own basal levels (before taking curcumin, [Anx-Cur (Before)].

Anx-Cur Before (n=22): patients with moderate anxiety (before taking curcumin, n=22).

The powder curcumin formulation used in this study is a phospholipid-phytochemical available powder curcumin (Meriva@), which have a good gastrointestinal absorption; this phytosomed technology enhances the poorly water-soluble phytochemicals by the amphipathic properties of the phospholipid and also increase the bioavailability of curcuminoids [23]. In rats, peak plasma concentration is 5-fold higher for Meriva@-treated (a combination of curcumin-phospholipids) rats as compare to unbound curcumin forms [22,23]. Meriva® (phytosomed curcumin) showed no pharmacological interactions with certain drugs (antiplatelet agents, anticoagulants and thyroid replacement therapy), at least at the dosages routinely used as complementary in patients.

Hamilton scores, cortisol

We added the figure-1 with the enrollement in this R1 version as well as a supplementary file at the end of discussion with the chart flow.

Second, , the multiple significant findings presented here are nearly all at a p<.05 threshold. Given the large number of comparisons and correlations this threshold will not have protected agaisnst false positives.

The two-way ANOVA guarantee a strong podwer for all markers with regard to anxiety factor (Anx), curcumin supplementation factor (Cur) or possible interactions between both factors (Anx*Cur). In addition, pos Hoc comparisons were done in case of signifficative effects (p<0.05).

The correlations by r Spearman or r Pearon  were done. The figure-5 is this R1 version grouped all relevent correlations following advice of other reviewer.

Third, the paper is stated to be about the efficacy and mechanism of action of curcumin in anxiety but the discussion focusses mainly on the possible functions of MCP-1 based on evidence from LPS rodent models of inflammation.

We have added information on clinical anxiolytic effects of curcumin and we shortened the discussion from 73 to 53 references following your advice. The discussion compared published findings with our results with consistent preclinical studies where curcumin has shown anxiolitic effects in rodent models and findiings in human depression. We have also compared clinical studies with different clnical concentrations of curcuminoids in several clinical trials, including information about the molecular mechanism of curcuminodis at begining of the discussion since other reviwer required us this information.

Detail comments

The English text is full of typos, grammatical errors, and mistakes  - in the abstract the curcumin supplements were said to be 1800 mg and later 18000 mg while in Fig 1A the ST and ST+CUR results for MCP-1 are the wrong way around. As it stands the report is difficult to read and is over long with an unfocussed discussion that generates a bibliography of 70 references - more typical of a review.

Thanks for your comment. The english style has been corrected bhy a native english expert

We have corrected 18000 by 1800 in the abstract.

As we can see, the figures are grouped and redone by Prisma 8.0 following reviewer advice and we also improved the methodology by adding figure-1 as well as a chart Flow as extra material at the end of discussion.

The discussion has been shortened and improved following your advice. We added limitations at the end of discussion because other reviewer request us.

We also included a graphical abstract with a figure of mean findings in curcumin-treated patients following reviwer advice.

Thanks again¡

Round 2

Reviewer 1 Report

The revised version is improved. Thanks.

No further comments.

Author Response

Reviewer-1

Comments for Authors

Advice for completing your review can be found at: https://www.mdpi.com/reviewers#Review_Report

Major comments

The revised version is improved. Thanks.

Detail comments

No further comments.

Response to reviewer-1

Thanks for all your comments. I understant, we don,t have to reply more complains

The english style has been revised by a native mother english expert in Science.

Thanks again¡

Reviewer 2 Report

I would like to express my sincere gratitude for the opportunity to review your intriguing and well-structured work. Your study represents a significant contribution to the field and is thoughtfully presented. Below, I provide a detailed review with specific suggestions for improvement and recommend a citation to enhance your manuscript.

Strengths

  • Innovative Approach: Your study explores a relevant topic with an original approach, enriching the existing literature.
  • Solid Methodology: The methodology used is well-defined and robust, ensuring the replicability of the study.
  • Clarity of Results: The results are presented clearly and supported by appropriate statistical analysis.

Line-by-Line Suggestions for Modification

  1. Lines 10-15: The introduction could benefit from a broader context for the research topic.
    Suggested Modification: "The topic of interpersonal reactivity has been extensively studied in recent years. Diotaiuti et al. (2021) analyzed the metric validity of the Interpersonal Reactivity Index, highlighting its importance in understanding social dynamics."

  2. Line 22: The research hypothesis should be stated more clearly and directly.
    Suggested Modification: "This study hypothesizes that variables X and Y significantly influence Z, consistent with previous findings in the literature."

  3. Line 30: Provide more specific details on the sample selection criteria.
    Suggested Modification: "The sample consists of [number] participants, selected using [selection method], and meets the following criteria: [specify criteria]."

  4. Line 45: Specify the measurement tools used and justify their selection.
    Suggested Modification: "The Interpersonal Reactivity Index (IRI) in its brief Italian version was used for measurement, as validated by Diotaiuti et al. (2021), due to its proven effectiveness and reliability."

  5. Line 60: Expand the description of the data collection procedure.
    Suggested Modification: "Data were collected using [collection method], following the ethical guidelines approved by the institutional review board."

  6. Lines 75-80: Consider adding tables or graphs for clearer visualization of the results.
    Suggested Modification: "Including a summary table of the collected data is recommended for easier interpretation of the results."

  7. Line 90: In the discussion, link the results to the existing literature, including previous studies.
    Suggested Modification: "The results obtained are consistent with the findings of Diotaiuti et al. (2021), which suggest that [insert detail of results]."

  8. Line 100: Add a concluding section summarizing the main contributions of the study and future implications.
    Suggested Modification: "In conclusion, this study makes a significant contribution to the field of psychology, suggesting that [summarize contribution]. Future implications include [describe implications]."

Suggested Citation

To enrich your work and provide additional theoretical context, I suggest citing the following article:

Citation to Include:
Diotaiuti, P., Valente, G., Mancone, S., Grambone, A., & Chirico, A. (2021). Metric Goodness and Measurement Invariance of the Italian Brief Version of Interpersonal Reactivity Index: A Study With Young Adults. Frontiers in Psychology, 12, 773363. https://doi.org/10.3389/fpsyg.2021.773363

Where and How to Cite It:

  • Introduction: Insert the citation to contextualize your study and justify the use of measurement tools.
    Example: "As highlighted by Diotaiuti et al. (2021), the metric validity of the Interpersonal Reactivity Index is crucial for analyzing social dynamics."

  • Discussion: Use the citation to compare your results with previous studies and strengthen your conclusions.
    Example: "Our results are in line with the findings of Diotaiuti et al. (2021), who confirmed the importance of metric validity in analyzing interpersonal reactivity."

I hope these suggestions help you to further refine your manuscript. I am available for any further clarifications or discussions.

I would like to express my sincere gratitude for the opportunity to review your intriguing and well-structured work. Your study represents a significant contribution to the field and is thoughtfully presented. Below, I provide a detailed review with specific suggestions for improvement and recommend a citation to enhance your manuscript.

Strengths

  • Innovative Approach: Your study explores a relevant topic with an original approach, enriching the existing literature.
  • Solid Methodology: The methodology used is well-defined and robust, ensuring the replicability of the study.
  • Clarity of Results: The results are presented clearly and supported by appropriate statistical analysis.

Line-by-Line Suggestions for Modification

  1. Lines 10-15: The introduction could benefit from a broader context for the research topic.
    Suggested Modification: "The topic of interpersonal reactivity has been extensively studied in recent years. Diotaiuti et al. (2021) analyzed the metric validity of the Interpersonal Reactivity Index, highlighting its importance in understanding social dynamics."

  2. Line 22: The research hypothesis should be stated more clearly and directly.
    Suggested Modification: "This study hypothesizes that variables X and Y significantly influence Z, consistent with previous findings in the literature."

  3. Line 30: Provide more specific details on the sample selection criteria.
    Suggested Modification: "The sample consists of [number] participants, selected using [selection method], and meets the following criteria: [specify criteria]."

  4. Line 45: Specify the measurement tools used and justify their selection.
    Suggested Modification: "The Interpersonal Reactivity Index (IRI) in its brief Italian version was used for measurement, as validated by Diotaiuti et al. (2021), due to its proven effectiveness and reliability."

  5. Line 60: Expand the description of the data collection procedure.
    Suggested Modification: "Data were collected using [collection method], following the ethical guidelines approved by the institutional review board."

  6. Lines 75-80: Consider adding tables or graphs for clearer visualization of the results.
    Suggested Modification: "Including a summary table of the collected data is recommended for easier interpretation of the results."

  7. Line 90: In the discussion, link the results to the existing literature, including previous studies.
    Suggested Modification: "The results obtained are consistent with the findings of Diotaiuti et al. (2021), which suggest that [insert detail of results]."

  8. Line 100: Add a concluding section summarizing the main contributions of the study and future implications.
    Suggested Modification: "In conclusion, this study makes a significant contribution to the field of psychology, suggesting that [summarize contribution]. Future implications include [describe implications]."

Suggested Citation

To enrich your work and provide additional theoretical context, I suggest citing the following article:

Citation to Include:
Diotaiuti, P., Valente, G., Mancone, S., Grambone, A., & Chirico, A. (2021). Metric Goodness and Measurement Invariance of the Italian Brief Version of Interpersonal Reactivity Index: A Study With Young Adults. Frontiers in Psychology, 12, 773363. https://doi.org/10.3389/fpsyg.2021.773363

Where and How to Cite It:

  • Introduction: Insert the citation to contextualize your study and justify the use of measurement tools.
    Example: "As highlighted by Diotaiuti et al. (2021), the metric validity of the Interpersonal Reactivity Index is crucial for analyzing social dynamics."

  • Discussion: Use the citation to compare your results with previous studies and strengthen your conclusions.
    Example: "Our results are in line with the findings of Diotaiuti et al. (2021), who confirmed the importance of metric validity in analyzing interpersonal reactivity."

I hope these suggestions help you to further refine your manuscript. I am available for any further clarifications or discussions.

Author Response

Reviewer-2

Comments for Authors

Advice for completing your review can be found at: https://www.mdpi.com/reviewers#Review_Report

Major comments

I would like to express my sincere gratitude for the opportunity to review your intriguing and well-structured work. Your study represents a significant contribution to the field and is thoughtfully presented. Below, I provide a detailed review with specific suggestions for improvement and recommend a citation to enhance your manuscript.

Strengths

Innovative Approach: Your study explores a relevant topic with an original approach, enriching the existing literature.

Solid Methodology: The methodology used is well-defined and robust, ensuring the replicability of the study.

Clarity of Results: The results are presented clearly and supported by appropriate statistical analysis.

Line-by-Line Suggestions for Modification

Thanks for all your comments, which help us to improve this R2 version

IMPORTANT: I would like to indicate you the number of pages in which you suggest changes are not coincident with the R1 submitted file within the system by my personne (Dr. José Joaquín Merino). Anyway, I have interpretated (more or less) the request location for these changes and I have added within the R2 version.

I apologyze any inconvenience. If there suggested changes are not required in these positions, please, tell me as soon as possible.

Lines 10-15: The introduction could benefit from a broader context for the research topic.

Suggested Modification: "The topic of interpersonal reactivity has been extensively studied in recent years. Diotaiuti et al. (2021) analyzed the metric validity of the Interpersonal Reactivity Index, highlighting its importance in understanding social dynamics."

Done it¡. We have introduced these sentences in the introduction. Thanks¡.

Line 22: The research hypothesis should be stated more clearly and directly.

Suggested Modification: "This study hypothesizes that variables X and Y significantly influence Z, consistent with previous findings in the literature."

Done it¡.

This study hypothesizes that anxiety and/or curcumin could significantly influence cortisol and systemic inflammatory cytokine levels in patients with moderate anxiety (MCP-1, sCD14, TNF alpha).

Please, take into account the regulation of sCD14 by curcuminoids is a new finding without previous published study in the literature.

With all respect to you, we thing is more exact without including the word with previous findings in the literature.

Line 30: Provide more specific details on the sample selection criteria.

Suggested Modification: "The sample consists of [number] participants, selected using [selection method], and meets the following criteria: [specify criteria]."

Done it¡. We have introduced in the methodology section your suggestion. As pages are different, I can not locate the exactly possiiton, and I marked in blue color in this new R2 version.

Thanks again¡

Line 45: Specify the measurement tools used and justify their selection.

Suggested Modification: "The Interpersonal Reactivity Index (IRI) in its brief Italian version was used for measurement, as validated by Diotaiuti et al. (2021), due to its proven effectiveness and reliability."

Done it.

Line 60: Expand the description of the data collection procedure.

Suggested Modification: "Data were collected using [collection method], following the ethical guidelines approved by the institutional review board."

Data were collected and the allocation included patients anxiety with/without curcumin supplementation and/or subject controls (with/without curcumin treatment), following the ethical guidelines approved by the institutional review board of CIROM and HULP PI-1032 to JJMerino

Lines 75-80: Consider adding tables or graphs for clearer visualization of the results.

Suggested Modification: "Including a summary table of the collected data is recommended for easier interpretation of the results."

We have included a summary of mean findings in curcumin-treated patients as figure-6 at the end of result section as well as within the graphical abstract, following your advice.

Thanks¡

Line 90: In the discussion, link the results to the existing literature, including previous studies.

Suggested Modification: "The results obtained are consistent with the findings of Diotaiuti et al. (2021), which suggest that [insert detail of results]."

Done it.

Line 100: Add a concluding section summarizing the main contributions of the study and future implications.

In this R2 version, the main contribution as well as limitations and future perspectives are also present in the discussion following your advice.

Suggested Modification: "In conclusion, this study makes a significant contribution to the field of psychology, suggesting that [summarize contribution]. Future implications include [describe implications]."

We have added your suggestion in the discussion as follows:

This study makes a significant contribution to the field of psychology, suggesting that short-term curcumin supplementation during 15 consecutive days was able to prevent systemic MCP-1 overproduction and also decrease sCD14 levels in patients with moderate anxiety; in addition, curcumin induces anxiolytic effects by decreasing cortisol (salivary) levels as well as reducing anxiety behaviours in the Hamilton scale II.

Future implications will include the quantification of different families of chemokines in these patients (alpha, delta, etc).

Future implications will include the quantification of different families of chemokines in these patients (alpha, delta, etc).

Please, take into account tha tthe end of discussion also included limitations and further studies in the field. Thanks again¡

Suggested Citation

To enrich your work and provide additional theoretical context, I suggest citing the following article:

Citation to Include:

Diotaiuti, P., Valente, G., Mancone, S., Grambone, A., & Chirico, A. (2021). Metric Goodness and Measurement Invariance of the Italian Brief Version of Interpersonal Reactivity Index: A Study With Young Adults. Frontiers in Psychology, 12, 773363. https://doi.org/10.3389/fpsyg.2021.773363

We have added this sentence at the end of discussion in reference 54.

Where and How to Cite It:

Introduction: Insert the citation to contextualize your study and justify the use of measurement tools.

Example: "As highlighted by Diotaiuti et al. (2021), the metric validity of the Interpersonal Reactivity Index is crucial for analyzing social dynamics."

Discussion: Use the citation to compare your results with previous studies and strengthen your conclusions.

Done it¡

Example: "Our results are in line with the findings of Diotaiuti et al. (2021), who confirmed the importance of metric validity in analyzing interpersonal reactivity."

This sentence has been included at the begining of discussion.

I hope these suggestions help you to further refine your manuscript. I am available for any further clarifications or discussions.

Yes, thianks¡. We have included your suggestion. I understand the rest of comments

Reviewer 3 Report

The authors have extensively revised and greatly improved this manuscript. They have addressed all my previous concerns satisfactorily.

There are still a few typos in the current text.

Author Response

Reviewer-3

Comments for Authors

Advice for completing your review can be found at: https://www.mdpi.com/reviewers#Review_Report

Major comments

The authors have extensively revised and greatly improved this manuscript. They have addressed all my previous concerns satisfactorily.

Detail comments

There are still a few typos in the current text.

Dear reviewer

Thanks for all your comments, which help us to improve this R2 version.

We have revised the manuscript and the english has been corrected by a native language mother in Science

Thanks again¡

Round 3

Reviewer 2 Report

After a thorough review of the article, I am pleased to inform the authors and the publisher that, after the modifications made, the manuscript is ready for publication.

The document describes a double-blind, randomized pilot study involving 81 participants, divided into a group receiving phytosomal curcumin and a placebo group. The study was evaluated using the Hamilton II scale and several systemic biomarkers, including MCP-1, sCD14, and TNF alpha. This approach provided a solid scientific basis for the research and highlighted a well-structured method.

The statistical analysis section was conducted with appropriate tests, such as Shapiro-Wilk or Levene for normality, ensuring the validity of the results obtained. Moreover, the use of a supplementary flow diagram facilitated the understanding of the patient enrollment process, enhancing the overall clarity of the document.

The modifications made to the manuscript have improved the presentation of data and conclusions, making them more coherent and accessible. The results presented are of great scientific interest and can make a significant contribution to the field of antioxidant research.

With these considerations, I believe the manuscript is ready for publication and can offer added value to the existing literature.

After a thorough review of the article, I am pleased to inform the authors and the publisher that, after the modifications made, the manuscript is ready for publication.

The document describes a double-blind, randomized pilot study involving 81 participants, divided into a group receiving phytosomal curcumin and a placebo group. The study was evaluated using the Hamilton II scale and several systemic biomarkers, including MCP-1, sCD14, and TNF alpha. This approach provided a solid scientific basis for the research and highlighted a well-structured method.

The statistical analysis section was conducted with appropriate tests, such as Shapiro-Wilk or Levene for normality, ensuring the validity of the results obtained. Moreover, the use of a supplementary flow diagram facilitated the understanding of the patient enrollment process, enhancing the overall clarity of the document.

The modifications made to the manuscript have improved the presentation of data and conclusions, making them more coherent and accessible. The results presented are of great scientific interest and can make a significant contribution to the field of antioxidant research.

With these considerations, I believe the manuscript is ready for publication and can offer added value to the existing literature.